# *Euglena*'s atypical respiratory chain adapts to the discoidal cristae and flexible metabolism

Zhaoxiang He[1,5], Mengchen Wu[1,5], Hongtao Tian[1,5], Liangdong Wang[2], Yiqi Hu[1], Fangzhu Han[1], Jiancang Zhou[3] ✉, Yong Wang [2,4] ✉ & Long Zhou [1] ✉

*Euglena gracilis*, a model organism of the eukaryotic supergroup Discoba harbouring also clinically important parasitic species, possesses diverse metabolic strategies and an atypical electron transport chain. While structures of the electron transport chain complexes and supercomplexes of most other eukaryotic clades have been reported, no similar structure is currently available for Discoba, limiting the understandings of its core metabolism and leaving a gap in the evolutionary tree of eukaryotic bioenergetics. Here, we report high-resolution cryo-EM structures of *Euglena*'s respirasome $I + III_2 + IV$ and supercomplex $III_2 + IV_2$. A previously unreported fatty acid synthesis domain locates on the tip of complex I's peripheral arm, providing a clear picture of its atypical subunit composition identified previously. Individual complexes are re-arranged in the respirasome to adapt to the non-uniform membrane curvature of the discoidal cristae. Furthermore, *Euglena*'s conformationally rigid complex I is deactivated by restricting ubiquinone's access to its substrate tunnel. Our findings provide structural insights for therapeutic developments against euglenozoan parasite infections.

The oxidative phosphorylation (OXPHOS) process converts energy in upstream metabolites including NADH and succinate into the universal energy currency ATP[1]. It usually involves four protein complexes (CI-CIV) constituting the electron transport chain (ETC) and a single complex V (CV) to drive ATP synthesis[2]. In addition to working individually, these complexes can assemble into supercomplexes (SCs) and megacomplexes (MCs), including SC $I + III_2$ [3–7], SC $III_2 + IV_{1/2}$ [8–11], the respirasome SC $I + III_2 + IV$ [12], as well as the recently reported ciliate SC $IV_2 + I + III_2 + II$ [13], mammalian MC $(I + III_2 + IV)_2$ [14] and the ciliate MC $IV_2 + (I + III_2 + II)_2$ [15]. To date, apart from the conserved core and accessory subunits, different numbers of lineage-specific accessory subunits have been identified for ETC complexes of eukaryotic supergroups including Opisthokonta (fungi and metazoan)[16–18], Archaeplastida

(algae and plants)[5,6,11] and Alveolata (ciliates and apicomplexans)[7,13,15]. They serve either stabilization, assembly or regulatory roles, or have enzymatic functions not related to canonical ETC functions[5,19,20]. Moreover, in situ cryo-electron tomographic (cryo-ET) studies of CV oligomerization and the recent cryo-electron microscopic (cryo-EM) structure of half-ring shaped ciliate megacomplexes all demonstrate that the OXPHOS system contributes to cristae morphogenesis in a lineage-specific manner[3,13,15,21].

*Euglena gracilis* is a free-living, flagellated single cell eukaryote and belongs to the phylum Euglenozoa which also harbors several important parasites including *Trypanosoma brucei* and *Leishmania donovani*[22]. Evolutionarily, *E. gracilis* represents the eukaryotic supergroup Discoba (originally belongs to the Excavata clade)[23,24] which still

[1]Department of Biophysics and Department of Critical Care Medicine of Sir Run Run Shaw Hospital, Zhejiang University School of Medicine, Hangzhou 310058, China. [2]College of Life Sciences, Zhejiang University, Hangzhou 310058, China. [3]Department of Critical Care Medicine, Sir Run Run Shaw Hospital, Zhejiang University School of Medicine, Hangzhou 310016, China. [4]The Provincial International Science and Technology Cooperation Base on Engineering Biology, International Campus of Zhejiang University, Haining 314400, China. [5]These authors contributed equally: Zhaoxiang He, Mengchen Wu, Hongtao Tian. ✉e-mail: jiancangzhou@zju.edu.cn; yongwang_isb@zju.edu.cn; longzhou@zju.edu.cn

lacks high-resolution ETC structure[25]. It possesses diverse metabolic strategies including aerobic respiration, anaerobic wax fermentation and photosynthesis via its secondary plastid[25]. *E. gracilis* mitochondria has stalked discoidal cristae, the edges of which are occupied by short helical segments of CV oligomers[22,26,27]. Mass spectroscopy and negative stain EM-based studies identify a previously unreported fatty acid synthesis (FAS) domain in both *E. gracilis* and *T. brucei* CIs (Eg-CI and Tb-CI), suggesting euglenozoan ETC is involved in the lipid metabolism apart from OXPHOS[28,29]. A helmet-like domain next to the cytochrome *c* (cyt *c*) binding site is also observed in Eg-CIV, likely contributing to the specific recognition for its endogenous cyt *c*[28,30]. Nevertheless, structures and functions of these atypical subunits and possible supercomplex organization of the euglenozoan ETC remain enigmatic.

Here we present high-resolution cryo-EM structures of the divergent Eg-SC I + III₂ + IV and Eg-SC III₂ + IV₂ to fill in a missing gap in the evolutionary tree of eukaryotic ETC. Our findings reveal details of the euglenozoan CI FAS and CIV helmet-like domains, shedding light on

their roles in *E. gracilis*'s anaerobic wax fermentation[31,32] and cyt *c* recognition[33]. Using molecular dynamics (MD) simulations, we demonstrate the distinctive supercomplex organizations that adapt to the negative membrane curvature of discoidal cristae[22,27] and lead to the exclusive existence of Eg-SC I + III₂ + IV but not the more widely conserved SC I + III₂[3,30]. By analyzing Eg-CI under different catalytic conditions, we propose an alternative coenzyme Q10 (CoQ) access-based active-deactive (A/D) transition mechanism for eukaryotic CIs without obvious conformational flexibility[6,7]. As the OXPHOS system becoming a promising target for anti-parasitic therapeutics, the current study could contribute to the treatments for infections such as trypanosomiasis and leishmaniasis[34,35].

## Results

### Overall structures of E. gracilis CI, SC I + III₂ + IV and SC III₂ + IV₂

Overall 2.81 Å and 3.06 Å cryo-EM reconstructions were obtained for Eg-SC I + III₂ + IV and Eg-SC III₂ + IV₂ respectively, using a mixed protein sample purified from *E. gracilis*'s mitochondria (Fig. 1a–c,

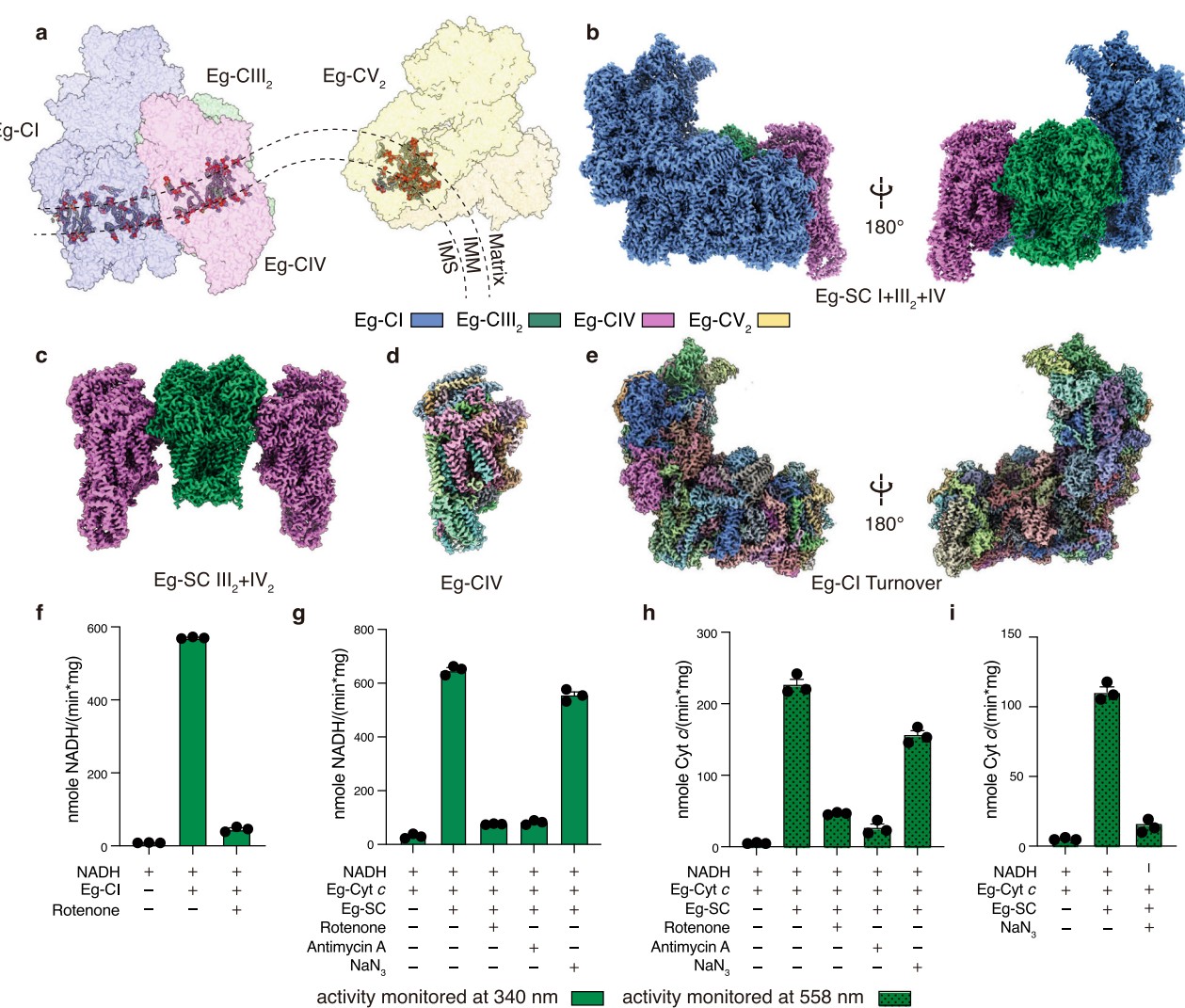

**Fig. 1 | Overall structures and functions of *E. gracilis*'s ETC. a** Representation of *E. gracilis*'s mitochondrial ETC on a discoid crista. Supercomplexes are shown in transparent surfaces colored by individual complexes (Eg-CV₂, PDB 6TDU); phospholipids are shown as spheres colored by elements. Eg-SC I + III₂ + IV (**b**) and Eg-SC III₂ + IV₂ (**c**) cryo-EM maps are colored by individual complexes as labeled. Eg-CIV (**d**) and Eg-CI (**e**) cryo-EM maps, which contain most of the *E. gracilis* specific subunits, are also colored by individual subunits. **f** NADH-DQ oxidoreductase activity of isolated Eg-CI in the absence or presence of 500 μM rotenone. NADH:O₂ oxidoreductase activities of Eg-SC I + III₂ + IV monitored by NADH oxidation at 340 nm (**g**), cyt *c* reduction at 558 nm (**h**) and cyt *c* oxidation also at 558 nm (**i**), in the absence or presence of 500 μM rotenone, 1 mM antimycin A or 100 mM NaN₃. Data are presented as mean values ± standard error of mean (SEM), *n* = 3 biologically independent activity experiments. Source data are provided as a Source Data file.

Supplementary Figs. 1–5, Supplementary Table 1 and Supplementary Movies 1 and 2). The divergent structures of Eg-CI with a matrix FAS domain and of Eg-CIV with an inter-membrane space (IMS) helmet domain were clearly revealed from the supercomplexes (Fig. 1b–d). To investigate the deactivation mechanism of Eg-CI, lauryl maltose neopentyl glycol (LMNG) solubilization was used to disrupt the supercomplexes (Supplementary Fig. 1). By pre-incubation with NADH only, with NADH and CoQ analog decylubiquinone (DQ), or at 37 °C for 20 min without any substrate, overall 3.03 Å, 2.87 Å and 3.11 Å reconstructions were obtained for Eg-CI under reduced, turnover and deactivated conditions respectively (Fig. 1e, Supplementary Figs. 6 and 7 and Supplementary Table 2). Recombinantly expressed Eg-cyt $c$ was used to confirm the electron transport activities of the protein sample (Fig. 1f–i and Supplementary Figs. 8 and 9a–c). The NADH oxidation and Eg-cyt $c$ reduction rates of digitonin-solubilized Eg-ETC complexes were both spectroscopically measured to be around 600 nmole/min/ mg protein, sensitive to inhibition by CI and CIII$_2$ inhibitors rotenone and antimycin A (Fig. 1g–i). 500 μM rotenone was used to fully inhibit Eg-CI as the measured half maximal inhibitory concentration (IC$_{50}$) is around 11 μM, much higher than the IC$_{50}$ of around 10 nM for rotenone inhibition of mammalian CI (Supplementary Fig. 9a, Supplementary Discussion)[4]. The same rotenone concentration was used previously to inhibit oxygen consumption and electron transport activity of purified *E. gracilis*'s respirasome[30]. CIV inhibitor sodium azide specifically inhibited Eg-cyt $c$ re-oxidation as demonstrated by the 558 nm kinetic curve (Fig. 1i). Local refinements were performed on different parts of above consensus maps before combined into composite maps used for modeling (Supplementary Figs. 2–7).

Models of Eg-CI, Eg-SC I + III$_2$ + IV and Eg-SC III$_2$ + IV$_2$ were built de novo according to densities using the *E. gracilis* genome and transcriptome[36] (Supplementary Fig. 10), of which 91%, 92% and 95% residues were at atomic level respectively (Supplementary Tables 3–5). Eg-CI comprises 59 subunits, of which 42 are conserved among all eukaryotes, four (NDUCA1–3 and NDUFX) are found in archaeplastidans and alveolates, one (NDUEG1, or NDTT2 in *T. thermophila* CI or Tt-CI) is conserved only in alveolates and 12 are unidentified Eg-specific subunits[7] (Supplementary Fig. 11 and Supplementary Tables 3, 5). Eg-CIII$_2$ and Eg-CIV each has 10 Eukaryota-conserved subunits, as well as one and nine Eg-specific subunits respectively[7] (Supplementary Fig. 12 and Supplementary Tables 3, 4). Altogether, Eg-SC I + III$_2$ + IV and Eg-SC III$_2$ + IV$_2$ have overall MWs of ~2.6 MDa and ~1.4 MDa respectively, about 1 MDa and 500 kDa larger than their opisthokont counterparts[10,12]. The membrane domains of individual complexes, as marked by modeled lipids and trans-membrane helices (TMHs), are arranged in a near planar architecture with slight negative curvature in Eg-SC I + III$_2$ + IV, indicating differential localization between ETC complexes and CV as in opisthokont mitochondria[21] (Fig. 1a and Supplementary Fig. 13).

## Eg-CI peripheral arm has a fatty acid synthesis domain and a split core subunit NDUFS1

An Eg-specific FAS domain formed by five subunits is found on the Eg-CI peripheral arm (PA) tip (Fig. 2a, b and Supplementary Figs. 14a–d, 15). Apart from subunit NDUEG1/NDUTT2/NDTB17 that is present in both Eg- and Tt-CI[7,28], the FAS domain also includes the NDUEG2 subunit and the NDUEG3-NDUEG5 heterodimer structurally homologous to the phosphoenolpyruvate carboxykinase (PEPCK)[37] and the medium-chain dehydrogenase/reductase (MDR) superfamily members[38], respectively (Fig. 2a, b). Presence of the FAS domain comprising some or all of the above described subunits has been confirmed by electrophoretic and mass spectroscopic studies in both Tb- and Eg-CIs[28,29]. Sequence BLASTs suggest that NDUEG3/NDTB2[28] and NDUEG5 are most similar to the NADH-dependent trans-2-enoyl-coenzyme A (CoA)/acyl carrier protein (ACP) reductases (TERs)[39], which reduce the enoyl thioester double bond thereby overcome the irreversible step in the β-oxidation pathway utilized by *E. gracilis* for

anaerobic wax fermentation[31]. A catalytically critical tyrosine (Tyr79 in 4WAS)[40] is conserved in NDUEG5 but not in NDUEG3 or the equivalent MDR-like subunits of *T. brucei* CI (Tb-CI)[29], making NDUEG5 the only possibly genuine NADH-dependent TER among euglenozoan CI FAS domains (Supplementary Figs. 14a–d, 15a–d). However, spectroscopic assays fail to detect any NADH- or NADPH-dependent reductive activity of crotonyl-CoA or 2-trans-dodecenoyl-CoA by Eg-CI sample, indicating that either NDUEG5 has alternative substrate specificity or its enzymatic activity is lost during co-evolution with euglenozoan CI (Supplementary Fig. 16a, b, d–g, Supplementary Discussion). PEPCK catalyzes the inter-conversion between oxaloacetate and phosphoenolpyruvate (PEP) using ATP or GTP as the phosphate donor, thereby overcomes the irreversible step in gluconeogenesis by $CO_2$ fixation[41]. Compared to soluble PEPCK structures from *E. coli* (Protein Data Bank (PDB) 1AQ2)[42] or *T. brucei* (PDB 1II2)[37], NDUEG2 has an incomplete C-terminal (CT) domain lacking two ATP binding sites thereby unlikely to be functional (Supplementary Fig. 15e, Supplementary Discussion).

Unlike the classic opisthokont CI with 14 core subunits, NDUFS1 and ND2, belonging to Eg-CI's PA and membrane arm (MA) respectively, are each split into two subunits, making a total of 16 core subunits in Eg-CI (Fig. 2c–f). Similar to Tt-CI ND2A and ND2B subunits which comprises non-metazoan ND2's first 10 and last 4 TMHs[7], Eg-ND2A and ND2B subunits together constitute its structural equivalent only that the eleventh TMH belongs to Eg-ND2A but not Eg-ND2B (Fig. 2c, d). It is widely recognized that post-endosymbiosis, mitochondrial genes are under constant pressure to relocate to the nucleus, despite the problematic import of highly hydrophobic proteins from cytosol back to mitochondria[43,44]. Unlike most cases, ND2 subunits are not among the seven proteins encoded by the streamlined mitochondrial genome of *E. gracilis*[45]. Similar to the case of subunit a in *Toxoplasma gondii* ATP synthase[46], splitting of Eg-ND2 between its eleventh and twelfth TMHs produces a less hydrophobic Eg-ND2B subunit short enough to avoid recognition by the signal recognition particle (SRP), which escorts protein to the endoplasmic reticulum rather than the mitochondria[47,48]. On the other hand, the way Tt-CI split its ND2 subunit produces a four-TMH Tt-ND2B subunit considered recognizable by SRP according to the published rules (Fig. 2c)[7,48]. The rich mitochondrial genome of *T. thermophila* contains ND2B gene among the other ~43 open reading frames[49], supporting the idea that such core subunit splitting is indeed driven by the mito-nuclear coevolution[50].

Apart from ND2, the hydrophilic core subunit NDUFS1, the largest subunit of CI across species, also splits into two subunits Eg-NDUFS1A and NDUFS1B, corresponding roughly to the NT [FeFe]-hydrogenase-like domain and the CT molybdopterin-containing enzyme-like domain of canonical NDUFS1[51] (Fig. 2e, f). Apart from the *E.coli* NuoCD subunit which splits into NDUFS2 and NDUFS3 in eukaryotic CI[52], splitting of hydrophilic core subunit in eukaryotic CI has never been reported to our knowledge. The evolutionary rationale for splitting nuclear-encoded NDUFS1 could relate to folding panelty differences between one large subunit and the heterodimer composing two smaller subunits after splitting[53]. In Eg-CI, NDUFS1A and NDUFS1B break off from each other at the central β-sheet in the Rossman fold-like subdomain II next to NDUFA6[51], such that the only anti-parallel β-strand belongs to NDUFS1A while the rest parallel β-strands belong to NDUFS1B (Fig. 2f).

Splitting of subunit NDUFS1 contributes to the association of the FAS domain onto Eg-CI PA via the connective accessory subunits NDUFA6 and NDUFS4 (Fig. 2g–i). The LYR motif protein NDUFA6 originally clusters with the ACP NDUAB1-α as part of the ferredoxin bridge in non-opisthokont CI[5–7,19]. Eg-NDUFA6 is highly augmented as it has a prominent, all α-helical CT extension reaching all the way towards NDUEG3 on the PA tip (Fig. 2h, i). It not only embeds the Eg-specific helical hairpin on NDUFS1A CT made possible by NDUFS1 splitting, but also clamps the CT vertical helix of NDUEG3 together with NDUFS1B.

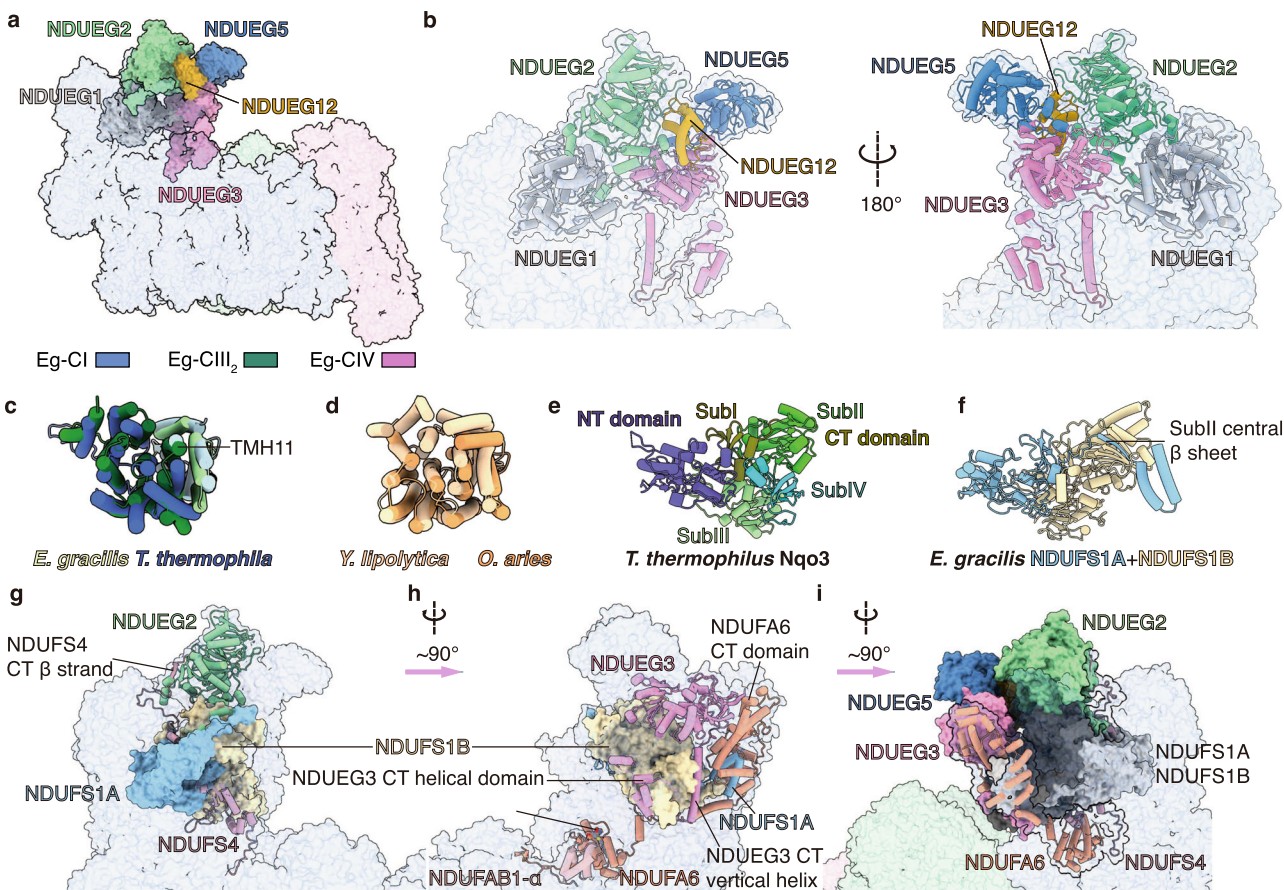

**Fig. 2 | The FAS domain and the split core subunits.** Overall structure (**a**) and zoom-ins (**b**) of the FAS domain within Eg-SC I + III₂ + IV. **c** Structural alignments of ND2 subunits between *E. gracilis* (this study, ND2A dark green, ND2B light green) and *T. thermophila* (PDB 7TGH, ND2A dark blue, ND2B light blue) are shown as super-imposed cartoons. **d** Structural alignments of ND2 subunits between *Y. lipolytica* (PDB 6RFR, light yellow) and *O. aries* (PDB 6ZKD, dark yellow) are also shown as a reference to (**c**). Structural comparison of *T. thermophilus* Nqo3 subunit

(PDB 2FUG) (**e**) and *E. gracilis* NDUFS1 subunit (this study) (**f**). The NT and CT domains of *T. thermophilus* Nqo3 are colored by blue and different shades of green marking the four subdomains. *E. gracilis* NDUFS1A and NDUFS1B are colored in light blue and yellow respectively. **g–i** Association of the FAS domain onto Eg-CI PA. The NDUFS1A and NDUFS1B are shown in surfaces and other subunits are shown in cartoons. Relative rotations of the point of view between individual panels are labeled by their directions and degrees.

This CT helix, not conserved in normal MDRs, is followed by a helical motif wrapping around the Eg-specific helical hairpin in NDUFS1B subdomain II, further tightening the association (Fig. 2h). Subunit NDUFS4 satisfies the evolutionary pressure for NDUFS1 stabilization by filling the groove between its NT and CT domains in both prokaryotic and eukaryotic CIs[19,52]. In Eg-CI, the Eg-specific NT loop of NDUFS1A covers this groove and turns it into an eyelet where NDUFS4's CT loop can only thread through NDUFS1A and NDUFS1B (Supplementary Fig. 15j). Such arrangement implicates that the accessory subunit NDUFS4 should participate early in the N module assembly together with core subunits NDUFS1A and NDUFS1B in Eg-CI, rather than serving as a checkpoint for the N module attachment by exchanging the assembly factor NUDFAF2 for NDUFA12[19,54]. Moreover, the CT β-strand of NDUFS4 integrates into the central β-sheet of NDUEG2's Rossman fold domain, thereby linking it to the NDUFS1A and NDUFS1B subunits (Fig. 2g). Together these two long accessory subunits NDUFS4 and NDUFA6 fixate the FAS domain on top of the core subunits NDUFS1A and NDUFS1B like two joint brackets nailed on both sides of such structural assembly (Fig. 2i, Supplementary Discussion).

## Architecture of Eg-SC I + III₂ + IV induces negative membrane curvature

Distinct interaction sites between CI-CIII₂ and CI-CIV are observed within the Eg-SC I + III₂ + IV compared to the existing mammalian (PDB 6QBX and 5J4Z)[4,12], plant (PDB 8BPX and 8E73)[5,6] and alveolate (PDB

7TGH)[7] respirasome and/or SC I + III₂ structures (Fig. 3). When aligning Eg-SC I + III₂ + IV to the mammalian respirasome by CI MA, Eg-CIII₂ exhibits a ~ 41 Å translation towards the CI PA as well as a ~ 10° rotation in the bilayer plane, establishing a major CI-CIII₂ interaction site in the matrix involving CI NDUFA9 and CIII₂ UQCRB from the proximal protomer (defined as the protomer with COB subunit closer to CI) (Fig. 3c and Supplementary Fig. 13c). The NADPH-containing CI accessory subunit NDUFA9, key to CI assembly and Q tunnel access during A/D transition, now has an additional role in respirasome assembly in our structure[19] (Fig. 3a and Supplementary Fig. 13i). Surprisingly, NDUCA1 belonging to the γ-carbonic anhydrase (γCA) domain extends a long NT loop that horizontally crosses Eg-CI MA and polarly contacts the only Eg-specific CIII₂ subunit UQCREG1 at this site (Supplementary Fig. 13e, Supplementary Discussion). The substantial relocation of CIII₂ fills the wedge between the CIII₂ distal protomer and the heel part of CI consistently observed in mammalian and plant SC I + III₂, while creates a different gap between CIII₂ proximal protomer and the CI MA (Fig. 3c and Supplementary Fig. 13a, b). Such gap, likely filled by membrane lipids, makes room for a minor CI-CIII₂ interaction site to exist in the matrix leaflet of the inner mitochondrial membrane (IMM), involving the Eg-specific CI subunit NDUEG9 and CIII₂ subunit UQCR9 (Fig. 3a, and Supplementary Fig. 13d). It is noteworthy that electron density for the bridging subunit NDUEG9 is discernible only in Eg-SC I + III₂ + IV but not Eg-CI, suggesting it is fully ordered only in the respirasome, reminiscent of the structural change of the ND5A's NT domain

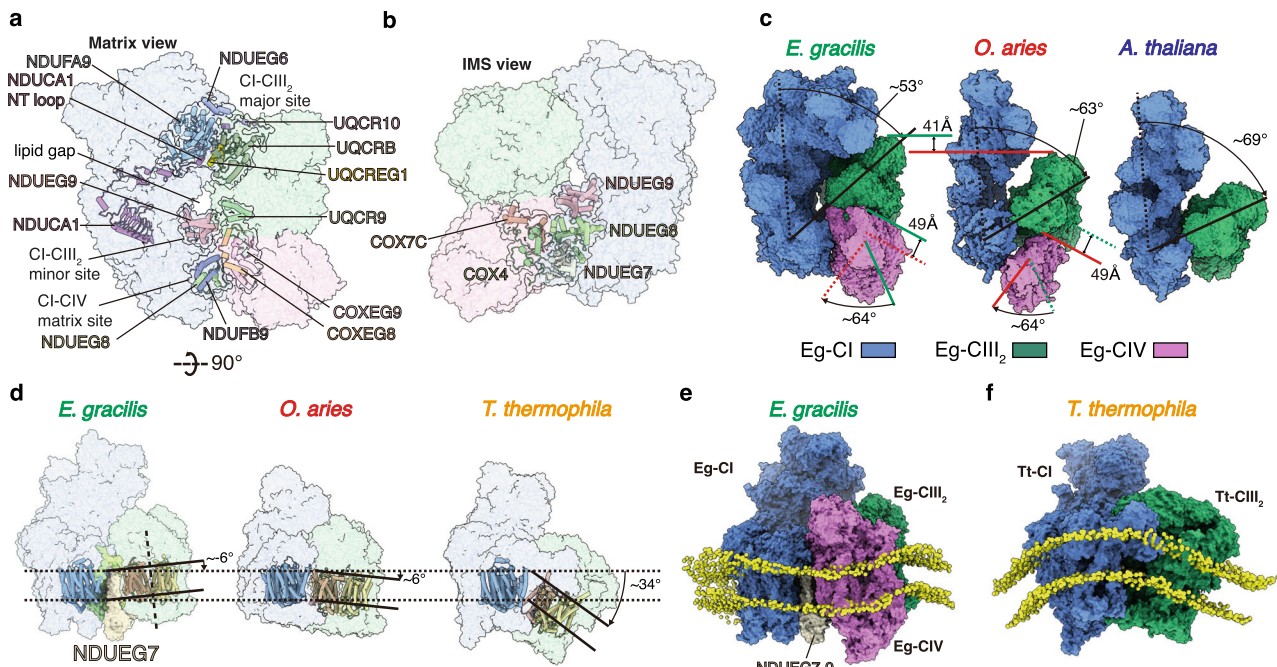

**Fig. 3 | Interaction sites within Eg-SC I + III₂ + IV.** Individual complexes are colored as labeled throughout this figure (CI: blue; CIII₂: green; CIV: magenta). Matrix (**a**) and IMS (**b**) view of interaction sites within Eg-SC I + III₂ + IV. Individual complexes are shown as colored transparent surfaces. Key subunits are shown as cartoons. **c** Comparisons of the SC I + III₂ + IV or SC I + III₂ from *E. gracilis*, *O. aries* (PDB 5J4Z) and *A. thaliana* (PDB 8BPX). Note that since Eg- and ovine respirasomes are shown side by side, positional comparisons of their CIII₂ and CIV are visualized by color-coded lines (green: Eg-; red: ovine) marking translational and rotational

positions of respective complexes. Translational distances and rotational angles are labeled. **d** Comparison of CI-CIII₂ angle in *E. gracilis*, *O. aries* and *T. thermophila* (PDB 7TGH), viewed from alongside the membrane plane. TMHs of ND5 and COB are shown in colored cartoons indicating the membrane curvatures, NDUEG7 is shown in solid surface. CG MD simulations of Eg-SC I + III₂ + IV (**e**) and Tt-SC I + III₂ (**f**) in lipid bilayers demonstrate that they induce negative and positive membrane curvatures respectively. Individual complexes are colored as in Figs. 1, 2. The CG lipid phosphate head groups are shown as yellow spheres.

between Tt-SC I + III₂ and Tt-MC IV₂ + (I + III₂ + II)₂[15] (Supplementary Fig. 10).

Due to CIII₂ relocation, canonical interactions between CIII₂ and CI MA toe are now mediated by CIV via one matrix and two IMS CI-CIV interaction sites (Fig. 3a, b). This explains why SC I + III₂, otherwise universally conserved in Eukaryota, is not observed for digitonin solubilized *E. gracilis* ETC, as deletion of CIV would break SC III₂ + IV's assoication to CI MA toe, leaving Eg-CIII₂ unstably attached only to the heel of Eg-CI[3,30]. The retraction of CIII₂ towards CI PA makes room for a ~64° rotation and ~49 Å translation of Eg-CIV towards the side of CI MA toe, thereby the front end of MA toe, a supercomplex interaction hotspot, is freed in Eg-SC I + III₂ + IV[12,13,15] (Fig. 3c). In the matrix, although the classic interaction partners of NDUFB9 and UQCRC1 are segregated by the CI-CIII₂ gap, their association is indirectly mediated by the Eg-specific matrix domain of CIV (Supplementary Fig. 13f, Supplementary Discussion). In the IMS, two CI-CIV interaction sites are observed, mediated respectively by Eg-specific CI accessory subunits NDUEG7 and NDUEG8 on the Q-tunnel side of MA (Fig. 3b and Supplementary Fig. 13g, h). NDUEG7, NDUEG8 and NDUEG9 are all TMH subunits with noticeable IMS domains but without matrix domain, lining up in a row and wedging between CI MA toe and SC CIII₂ + CIV in Eg-SC I + III₂ + IV (Fig. 3b and Supplementary Fig. 13i). In this way, the IMS ends of CI MA and SC III₂ + IV are pushed apart, while their matrix ends remain adjacent due to the interaction between NDUFB9 and COXEG8 of Eg-CIV's matrix domain (Supplementary Discussion). Thereby, TMHs from CI and SC III₂ + IV exhibits a negative membrane curvature of ~−6° when viewed from CI MA's distal end. This is in contrast to the increasing positive membrane curvatures observed in mammalian (−6°)[4], plant (−14°)[5,6] and alveolate (−34°)[7] SC I + III₂ (Fig. 3d).

To examine the membrane reorganization around the supercomplex, we performed coarse-grained (CG) MD simulations of the Eg

SC I + III₂ + IV. The results suggest that it indeed induced a negative membrane curvaton in its immediate surroundings (Fig. 3e and Supplementary Movie 5). In contrast, the CG MD simulations of the *T. thermophila* SC I + III₂ (PDB 7TGH) showed that it induced a positive curvature (Fig. 3f and Supplementary Movie 6)[7]. This distinctive negative CI-CIII₂ angle and induced membrane curvature, which has not been reported elsewhere to our knowledge, likely accommodates the region between the positively curved rims and the relatively flat center within the discoidal cristae of euglenozoans, placing Eg-SC I + III₂ + IV in close proximity to the oligomers of Eg-CV[22,27] (Fig. 1a).

### Active-deactive transition of Eg-CI

Standard CI A/D transition assays using thermal deactivation at 37 °C were performed with purified respirasome/SC I + III₂, CI and isolated mitochondrial membranes of *Sus scrofa* (pig), *T. thermophila* and *E. gracilis* (Fig. 4a–d, Supplementary Fig. 17 and Supplementary Fig. 9d–g). The *E. gracilis* sample can be thermally deactivated to certain degrees in the presence of *N*-ethylmaleimide (NEM) which stabilizes CI's deactive state, although less significantly than that of *S. scrofa* (Fig. 4a–d and Supplementary Fig. 17). Interesting, unlike *S. scrofa* CI and respirasome whose deactivation can be rescued by incubation with a small amount of re-activating NADH, Eg-SC I + III₂ + IV's susceptibility to NEM is not changed upon NADH re-activation (Fig. 4a–d and Supplementary Fig. 17), in line with the A/D transition properties of the recently reported vascular plant SC I + III₂[6]. 3D classification focused on the CI PA region of Eg-SC I + III₂ + IV particles pre-aligned by their CI MA revealed two major classes with only minute change in the PA-MA angle (Fig. 4e). Close inspections of the NDUFS2 β1-β2 loop, NDUFS7 loop and ND1 TMH5-6 loop shaping the Q tunnel show that they are in the retracted or ordered conformations in both classes, in line with the long and narrow form of Q tunnel found in the closed state of mammalian CI[16] (Fig. 4f, g and Supplementary

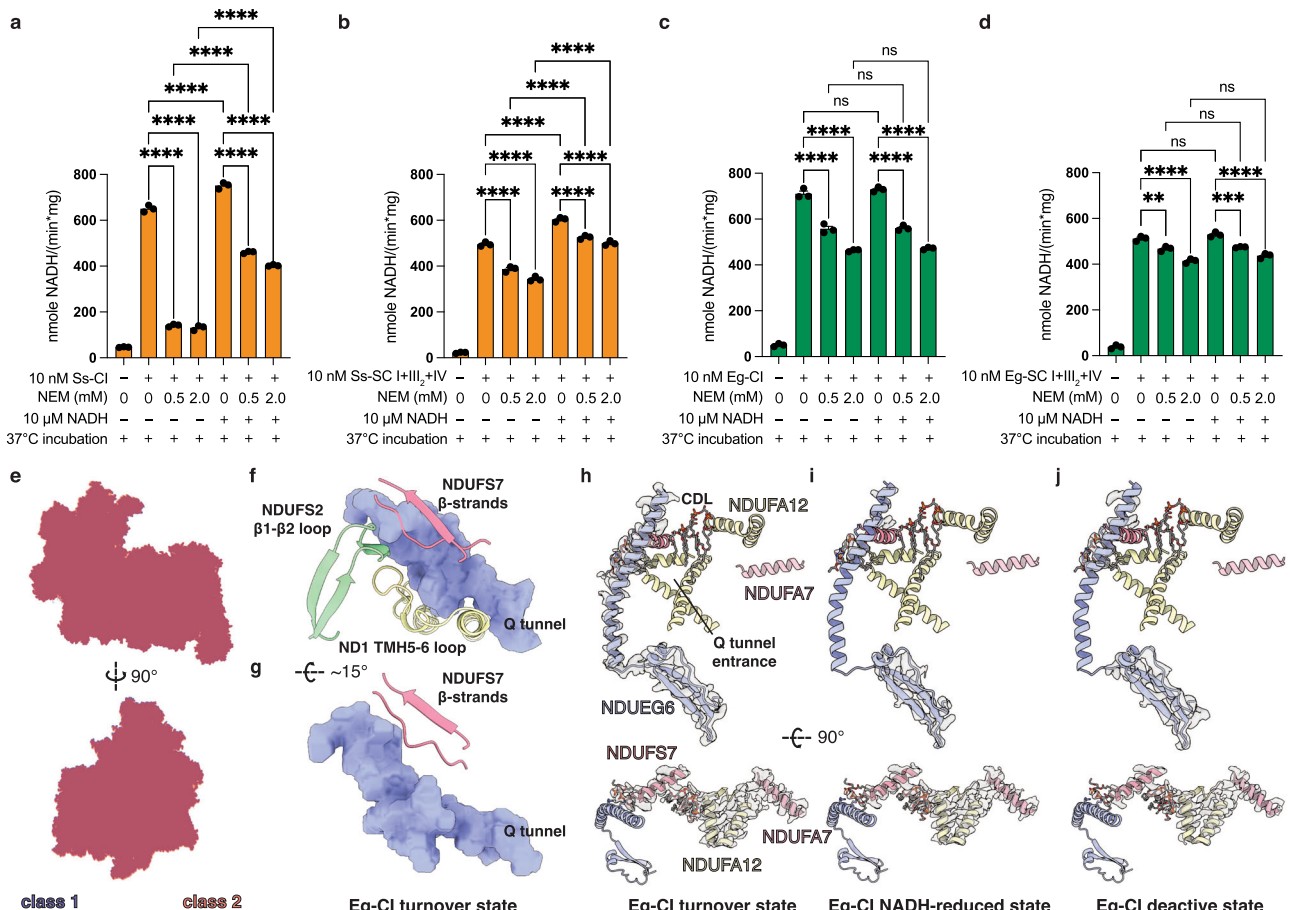

**Fig. 4 | Eg-CI's A/D transition.** Thermal deactivation assays of *S. scrofa* CI (**a**), *S. scrofa* SC I + III$_2$ + IV (**b**), *E. gracilis* CI (**c**) and *E. gracilis* SC I + III$_2$ + IV (**d**), measured by NADH oxidation at 340 nm. The assays are performed in the presence of the indicated concentrations of NEM, with or without 10 μM NADH re-activation. Data are presented as mean values ± standard error of mean (SEM), n = 3 biologically independent activity experiments. Statistical analysis is performed with one-way ANOVA with Tukey's multiple comparisons test. **, $p < 0.01$; ***, $p < 0.001$; ****, $p < 0.0001$. ns, not statistically significant ($P > 0.05$). Please note that the exact $p$ value is not provided by Prism when <0.0001 (****). For (**c**), the three comparison tests giving 'ns', from left to right, have $p$ values of 0.5471, 0.9996 and 0.9602, respectively. For (**d**), the comparison tests giving '***', '**' and 'ns', from left to right,

have $p$ values of 0.0002, 0.0017, 0.3271, 0.9472 and 0.1270, respectively. **e** Structural alignment of class 1 (purple) and class 2 (salmon) Eg-SC I + III$_2$ + IV, shown as flat surfaces. Note that the two classes do not exhibit major global conformational difference. **f, g** Local conformations of Q tunnel loops in the Eg-CI turnover structures. The Q tunnel is shown as light blue surfaces. Slight rotation exists between the two panels to better illustrate Q tunnel shaping capability of NDUFS7 β strands. Densities for TMH$^{NDUEG6}$, amphipathic helices of NDUFS7, NDUFA7 and NDUFA12, as well as key cardiolipins, are shown as transparent surfaces for Eg-CI turnover (**h**), NADH-reduced (**i**) and deactivated (**j**) states. Source data are provided as a Source Data file.

Fig. 18), although no CoQ density is identified in any of our Eg-SC I + III$_2$ + IV classes (Supplementary Fig. 18i, j). The key TMH3$^{ND6}$ is a straight α-helix without π-bulge indicating that the central hydrophilic axis in CI MA is connected in both classes[16,55] (Supplementary Fig. 18e).

The existence of distinct open and close states is not a prerequisite for conformational changes in the Q tunnel loops or ND6 TMH3, as *Yarrowia lipolytica* and *Chaetomium thermophilum* CIs do not show significant PA-MA angle change but do exhibit different local conformations in the above regions[18,56]. On the other hand, *T. thermophila*, *E. gracilis* and plant CIs with the ferredoxin bridge connecting the two arms have not only rigid PA-MA angle but also fixed Q tunnel loop and TMH3$^{ND6}$ conformations[5-7] (Supplementary Fig. 13m). Although Tt-CI cannot be deactivated thermally, how does the latter two CIs enter deactive state without noticeable global or local conformational flexibility[6,7] (Fig. 4c, d and Supplementary Fig. 17c, f, i, j)? To answer the question, we compared LMNG solubilized Eg-CI structures under NADH-reduced, turnover and thermally deactivated conditions. Density for bound NADH is found in the turnover and reduced maps, while resolution limits confident assignment of the tube-like density observed at the Q tunnel entrance as DQ (Supplementary

Fig. 18f–h). Notably, the sigmoid TMH of the Eg-specific subunit NDUEG6, positioned next to the conserved triangular α-helical pore of ND1 which marks the entrance of Q tunnel, is ordered only in the turnover map (Fig. 4h–j). Disordering or un-winding of this TMH and the β-strands on its CT in the deactivated map could lead to its dispositioning and subsequent impaired access to the Q tunnel. Moreover, several lateral amphipathic helices from subunits including NDUFS7, NDUFA7 and NDUFA12 exist in this region, inducing local thinning of the IMM and assist CoQ access to the Q tunnel[54] (Supplementary Discussion). In the deactivated Eg-CI map, density for a cardiolipin wedging in between TMH$^{NDUEG6}$ and the amphipathic helix of NDUFS7 is highly fragmented while two nearby cardiolipins between NDUFA12 and NDUFA7 lose their tail densities, indicating substantial loss of the local membrane thinning effect (Fig. 4h–j).

Taken together, above structural changes near the Q tunnel entrance upon thermal deactivation could collectively lead to impaired CoQ access to the Q tunnel thereby decreased CI activity. This represents a different mechanism for CI deactivation from eukaryotic lineages with the ferredoxin bridge, in line with their less significant activity loss upon thermal deactivation. It also explains why

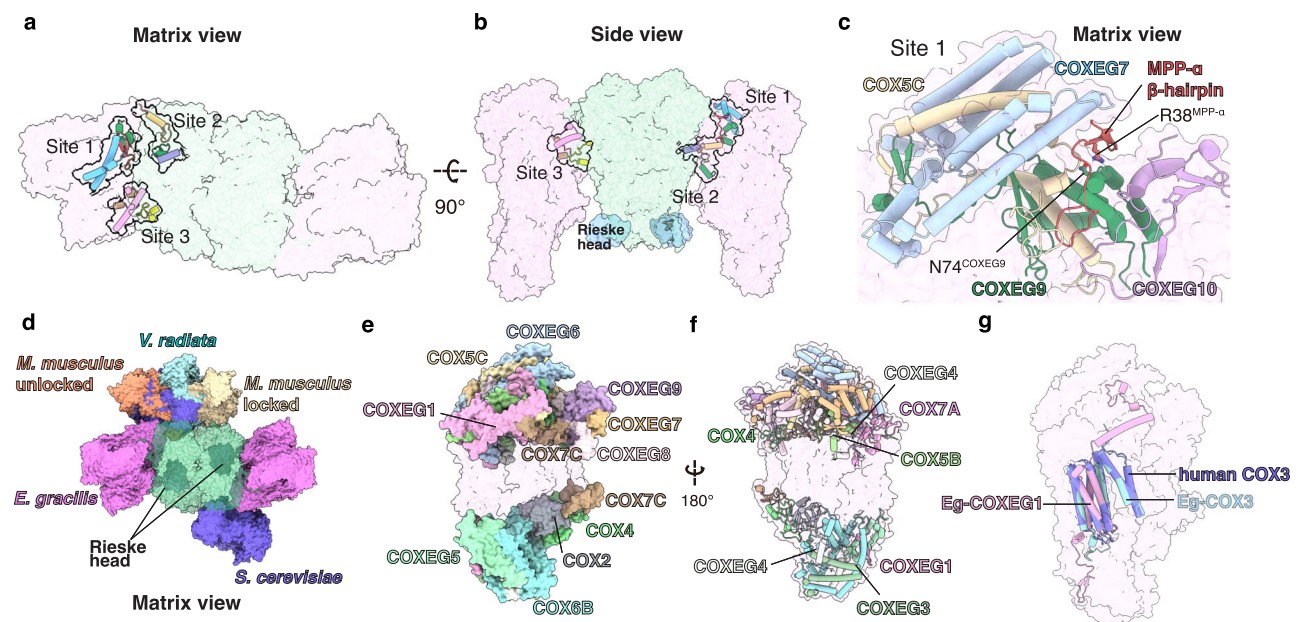

**Fig. 5 | Interaction sites within Eg-SC III$_2$ + IV$_2$.** Matrix (**a**) and side (**b**) views of Eg-SC III$_2$ + IV$_2$ interaction sites 1–3 with individual complexes shown as colored transparent surfaces. Rieske heads are shown as light blue solid surfaces. **c** Zoom-ins of Eg-CIII$_2$ + CIV$_2$ interaction site 1. Key residues forming polar contacts are shown as sticks and colored by elements. In site 1 MPP-α' β hairpin (red) wedges into a cleft on CIV surface. **d** Alignment of SC III$_2$ + IV$_{1/2}$ from different species by the COB dimer viewed from the matrix side, including, *M. musculus* unlocked (PDB 7O3C), *M. musculus* locked (PDB 7O37), *V. radiata* (PDB 7JRP), *S. cerevisiae* (PDB 6T0B) and *E. gracilis* (this study). The matrix domain and the IMS helmet-like domain of Eg-CIV are shown as colored solid surfaces (**e**) and cartoons (**f**) in the context of the Eg-CIV. **g** Structural alignment of Eg-COX3 and Eg-COXEG1 to human COX3 (slate, PDB 5Z62) demonstrating that TMHs from the two Eg subunits together constitute a canonical seven-TMH COX3 subunit found in mammals.

Eg-CI's NEM susceptibility is indifferent to pre-activation by NADH, as only interaction with DQ can relief the deactivated state by ordering the TMH$^{NDUEG6}$ and key lipids (Fig. 4a–d, h-j and Supplementary Fig. 17). On the other hand, how the conformationally rigid Eg-CI manages to complete its catalytic turnover, which requires CoQ to enter and exit its Q tunnel, awaits more in-depth mechanistic investigation (Supplementary Fig. 18l–n).

## Architecture of Eg-SC III$_2$ + IV$_2$ and its implications on CIII$_2$ functionality

Eg-SC III$_2$ + IV$_2$ is a ~ 1.4 MDa supercomplex with C2 symmetry where each CIII$_2$ protomer bound one copy of Eg-CIV, similar to the SC III$_2$ + IV$_2$ architectures of *Saccharomyces cerevisiae* and actinobacteria[8,9] (Figs. 1c, 5a, b). While the structure of Eg-CIII$_2$ is mostly conserved compared to those of opisthokonts and archaeplastidans, structure of Eg-CIV is quite divergent as it has nine Eg-specific subunits organizing into a matrix domain and an IMS helmet domain not present in its counterparts in any other species including the highly divergent Tt-CIV$_2$[7,11] (Fig. 5e, f). Notably, core subunit Eg-COX3 lacks three TMHs compared to its mammalian counterpart possibly due to the evolutionary pressure of mitochondrial genome streamlining[45]. The leftover space is occupied by TMHs of the nuclear-encoded subunit COXEG1, which introduces additional NT and CT extensions as parts of the matrix and IMS helmet domains respectively (Fig. 5g). Functional purpose of the Eg-specific matrix domain is apparently maintaining the CIII$_2$-CIV association solely via three matrix interactions sites (Fig. 5a–c and Supplementary Fig. 13c, j, k, Supplementary Discussion). Among them, the Eg-specific loop and β-hairpin on the NT of CIII$_2$'s UQCRC2/MPP-α wedge into a cleft on CIV surface, formed by the matrix parts of subunits COXEG7, COXEG9, COXEG10 and COX5C (Fig. 5c). It is noteworthy that the NT part of UQCRC2 is ordered only when interacting with nearby CIV, as its density is absent in the distal copy of UQCRC1 in Eg-SC I + III$_2$ + IV (Supplementary Fig. 10). This is similar to the ordering of subunit NDUEG9 in Eg-SC

I + III$_2$ + IV or the ND5A NT domain in Tt-MC IV$_2$ + (I + III$_2$ + II)$_2$ but happens between CIII$_2$ and CIV[15] (Supplementary Fig. 10).

Lack of IMM or IMS interactions sites between Eg-CIII$_2$ and Eg-CIV creates a prominent cleft between the two complexes, a feature not observed in other eukaryotic SC III$_2$ + IV$_{1/2}$[9–11] (Fig. 5b). When aligning their structures from *S. cerevisiae* (PDB 6T0B), *Vigna radiata* (PDB 7JRP) and *Mus musculus* (PDB 7O37 and 7O3C) to Eg-SC III$_2$ + IV$_2$ by the COB dimer of CIII$_2$, CIV position in Eg-SC III$_2$ + IV$_2$ distinguish itself from others as it approaches CIII$_2$ from the inter-COB groove rather than from the periphery of COB where the single TMHs of UQCRQ and CYC1 sit (Fig. 5d). Therefore unlike in other species, UQCRFS1 Rieske heads sit next to the CIII$_2$-CIV interface in Eg-SC III$_2$ + IV$_2$ such that any close CIII$_2$-CIV contact in the IMS would inevitably impose steric barrier on their swinging movements during the Q-cycle based electron transport[57] (Fig. 5d). To avoid such restriction, assoiation between Eg-CIII$_2$ and Eg-CIV is dominated by interactions in the matrix, while inter-complex cleft in the IMM and IMS distance the CIII$_2$ Rieske head from CIV (Fig. 5b). Indeed, NADH:O$_2$ oxidoreductase activity assays and 3DVA together indicate freely rotating and functional Rieske heads for both CIII$_2$ protomers in Eg-SC I + III$_2$ + IV and Eg-SC III$_2$ + IV$_2$ (Supplementary Movies 3, 4).

## An alternative mechanism for specific cytochrome c recognition
In line with previous reports[33], spectroscopic assays demonstrate that mammlian cyt *c* can be reduced by Eg-CIII$_2$ but cannot pass on electron to Eg-CIV, similar to the case of *T. thermophila* CIV$_2$ and cyt *c* due to charge-swapped interface between the two[7] (Fig. 6a, b and Supplementary Fig. 19). However Eg-CIV's cyt *c* binding site displays mainly negative surfacial electrostatics similar to that of mammalian CIV, while the CIV contacting surfaces of equine (PDB 1HRC)[58] and *E. gracilis* (predicted by AlphaFold2) cyt *c* structures are both positively charged (Fig. 6c). To clarify the mechanism for Eg-CIV's specific cyt *c* recognition, we aligned the Eg-CIV structure with the equine cyt *c* bound bovine CIV structure (PDB 5IY5)[59] using subunit COX1 to visualize

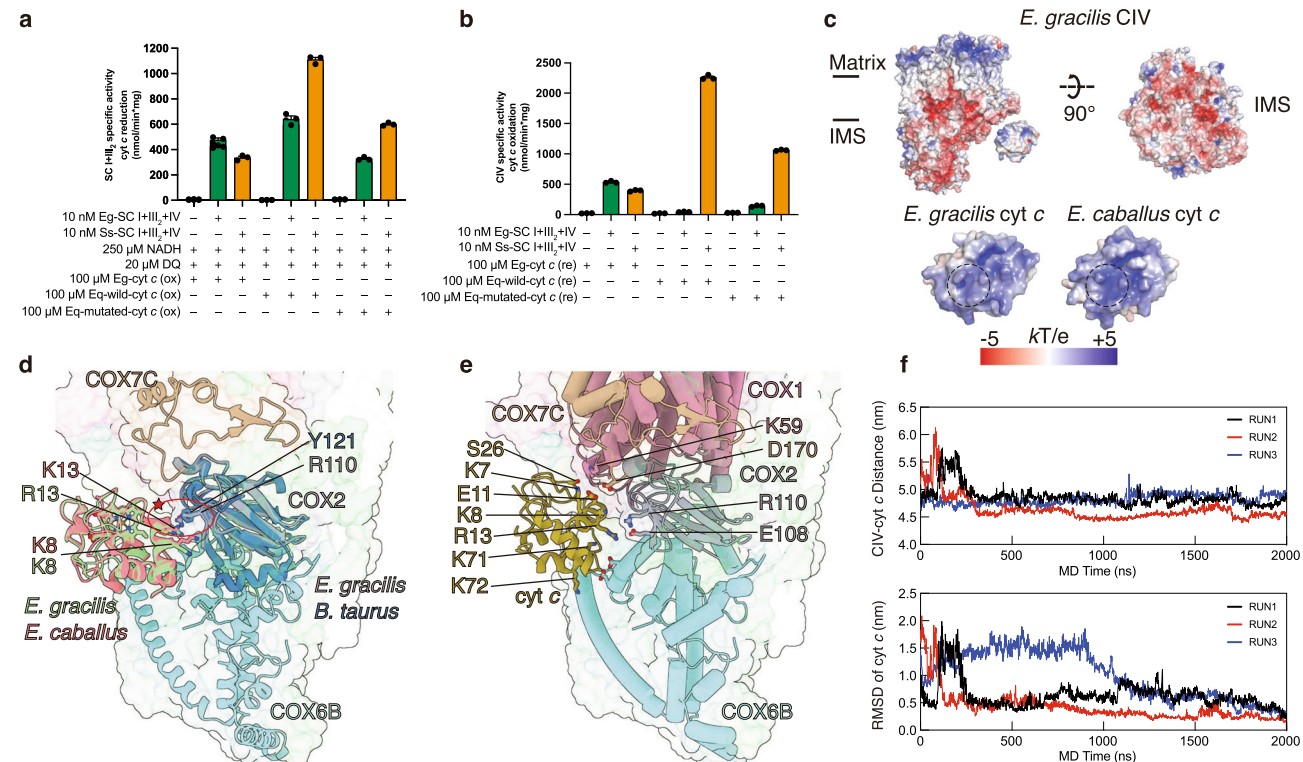

**Fig. 6 | Functional and structural divergence of Eg-CIV and cyt *c*.** Spectroscopic assays of cyt *c* reduction activities (**a**) and cyt *c* oxidation activities (**b**) of *E. gracilis* (green) and *S. scrofa* (orange) SC I + III$_2$ + IV, monitored at 558 or 550 nm for Eg- or equine cyt *c* respectively as indicated. Data are presented as mean values ± standard error of mean (SEM), *n* = 3 biologically independent activity experiments. **c** Surface electrostatic potentials of Eg-CIV, equine cyt *c* (PDB 1HRC) and Eg-cyt *c* (UniProt: P00076, AlphaFold predicted structure). Electrostatic potentials are calculated using the APBS plugin of PyMOL colored by a symmetric color ramp from −5 kT/e (red) to +5 kT/e (blue). **d** Eg-CIV and Eg-cyt *c* aligned to the mammalian CIV-cyt *c*

complex structure (PDB 5IY5). The star indicates electrostatic repulsion between Eg-CIV and Eg-cyt *c* if adopting the same association orientation as mammalian CIV-cyt *c*. Key residues are shown as sticks and colored by elements. **e** MD simulation (Run 1) of Eg-CIV+Eg-cyt *c* with potential key interacting residues shown as sticks and colored by elements. **f** MD trajectories (Run 1–3) of the center-of-mass distance between CIV and cyt *c* (upper panel) and the root-mean-square deviation (RMSD) of cyt *c* relative to the last frame of the simulation (lower panel) after aligning the entire CIV-cyt *c* complex with CIV during the 2000 ns MD simulation run. Source data are provided as a Source Data file.

interactions between Eg-CIV and a mammalian cyt *c* docked at its proposed binding site (Fig. 6d). Although most salt-bridging residues between COX2 and cyt *c* are conserved in both species, the substitution of mammalian Tyr121$^{COX2}$ with Eg-Arg110$^{COX2}$ creates potential electrostatic replusion to mammalian cyt *c* residues Lys9 and/or Lys14 (Fig. 6d), suggesting that mammalian cyt *c* cannot adopt its usual binding orientation when approching Eg-CIV.

However these two positively charged residues are conserved between mammalian and *E. gracilis* cyt *c*. To understand how Eg-cyt *c* binds Eg-CIV despite potential electrostatic repulsion at the interface, we performed explicit solvent all-atom MD simulations in quadruplicates, totaling 7 μs (Fig. 6e, f and Supplementary Fig. 19a–c). The initial Eg-CIV+Eg-cyt *c* structure was constructed using the mamalian CIV+cyt *c* structure (PDB 5IY5) as a template[59]. Three out of four of these simulations revealed stable binding of Eg-cyt *c*, positioning its heme *c* within direct electron transfer distance to COX2's dinuclear Cu$_A$ (Fig. 6e, f and Supplementary Fig. 19a, Supplementary Discussion). A close examination of the resulting structures showed that Eg-cyt *c* had rotated away from COX2's β-sheet cluster, alleviating Lys9-Arg14 induced electrostatic repulsion. Meanwhile, additional electrostatic interactions were established between Eg-cyt *c* and COX7C's CT loop, COX1's TMH1-2 and TMH3-4 loops and two perpendicular helices of COX6B belonging to Eg-CIV's IMS helmet domain, fixing Eg-cyt *c* in this new electron donating orientation (Fig. 6e). Although our simulations offer a feasible model for the binding mechanism, we acknowledge the difficulties in precisely quantifying binding affinities and comparing binding site stabilities given the limited timescales achievable with MD

and the probable inadequate conformational sampling. These limitations present opportunities for further investigation to build upon these preliminary findings.

## Discussion

Here we present structures of SC I + III$_2$ + IV and SC III$_2$ + IV$_2$ from the model organism *E. gracilis* of the Discoba supergroup (Fig. 1a–c). A classic puzzle in the field of ETC diversity is why a substaintial amount of accessory subunits are gained post-endosymbiosis without significant gain of function upon α-proteobacterial ETC (See Supplementary Tables 6–8 for an updated summary of ETC subunit conservation across lineages)[60,61]. This has been vividly examplifed by the highly divergent *T. thermophila* ETC with a total of 78 ciliatespecific subunits identified (20 for CI, 11 for CII, 2 for CIII$_2$ and 45 for CIV$_2$) without bringing along detectable novel function[7,15]. Another well-known example is the γCA domain universally present in nonopisthokont eukaryotic CIs, whose CO$_2$ hydration activity is retained or not is still under debate[5–7]. In Eg-CI and Tt-CI, none of the catalytically essential Zn$^{2+}$ coordinating histidine residues is conserved, nor do we observe any Zn$^{2+}$ or additional substrate density at the catalytic interfaces, making these γCA domains unlikely to be functional (Supplementary Fig. 14e and Supplementary Fig. 20)[5,62,63]. Efforts are made to assign adaptive functions to CI-bound γCA domain as a consequence of positive selection, however such proposals remain speculative in the absence of direct activity measurements.

In *E. gracilis* ETC, lineage-specific FAS and helmet domains are found in Eg-CI and CIV respectively, bringing a total of 22 previously

unidentified subunits. Similar adaptive narrative can be applied here, as under anaerobic conditions, *E. gracilis* uses rhodoquinone (RQ)-based ETC to turn fumarate into succinate by CII, using electrons coming from CI. In this process, PEPCK supplies the terminal electron acceptor fumarate by producing its precursor oxaloacetate from PEP, while TER consumes the end product succinate via the synthesis of odd-chain fatty acids during wax fermentation[31,64]. Integration of rate-limiting enzymes PEPCK and TER into CI could at least localize them to the cristae and spatially close to the anaerobic ETC CI-RQ-CII, for possible kinetic advantages[65]. However, the hypothesis is again challenged by the lack of detectable enzymatic activity for the FAS subunits (Supplementary Fig. 16), indicating that even if above scenario did happen during evolution, the FAS domain later became a vestige with its function compensated by the appearance of soluble TER isoforms or by the switch of the enoyl-CoA reducing enzyme from TER to acyl-CoA dehydrogenase (ACD)[31,32].

On the other hand, it has long been suggested that due to the small effective population size of the mitochondrial genome, non-adaptive, stochastic processes may lead to mutation accumulation that induces secondary selection of the nuclear-encoded mitochondrial proteins for counter-adaptation[50,66]. Alternatively, the evolutionary pressure of streamlining the mitochondrial genome causes simplification of its genes, also asking for compensation of structural integrity via mitonuclear coevolution[43]. Both processes lead to acquisition of additional elements and growing complexity, exemplified by the structural patching of mitochondrial ribosomes initiated by mitochondrial rRNA simplification[67,68]. Although the FAS domain seems unrelated to mitochondria-encoded subunits, these theories are applicable to the acquisition of Eg-specific CIV domains. Nuclear-encoded subunit COXEG1, patching the void space of absent COX3 TMHs, introduces long NT and CT extensions that encourages recruitment of additional matrix and IMS domains to minimize unwanted exposure of peptide backbone to solvents[69]. In this scenario, the IMS helmet domain is not acquired under selective pressure to better bind cyt *c*, in accordance with our results from MD simulations.

It is worth mentioning that adaptive selection also plays a role in the complexification observed for *E. gracilis* ETC, relating mostly to the rearrangement of respirasome architecture in active adaptation to discobans' discoidal cristae (Fig. 3c, d). The wedging subunits NDUEG7, NDUEG8 and NDUEG9 are clearly gained to introduce negative curvature between CI and SC III$_2$ + IV within Eg-respirasome. Moreover, although the γCA domain may be gained non-adaptively, in Eg-respirasome a specific NDUCA1 NT loop participates in the CI-CIII$_2$ interface, indicating that new function could emerge upon pre-existing accessory subunit under mutual shaping between OXPHOS complexes and mitochondrial cristae. Similar cases have been reported for ciliates and apicomplexans displaying distinct cristae morphologies, emphasizing the importance of adaptive selection in the evolution of respiratory chain[7,15,46,60,70,71].

Involvement of subunit NDUFA6 in the association of the FAS domain implicates how it is assembled onto Eg-CI. In the canonical assembly process of opisthokont CI, the N module attaches onto a pre-assembled Q/P subcomplex after a NDUFS6 checkpoint exchanging the assembly factor NDUFAF2 for subunit NDUFA12[72,73]. Close examination of the relative positions among NDUFA9, NDUFA6 and the FAS domain subunit NDUEG3 indicate that NDUEG3's CT vertical helix is spatially adjacent to NDUFA9's central β sheet, so that NDUFA6's helix cannot squeeze through in between with the other two subunits pre-assembled (Supplementary Fig. 21a). NDUFA9 is a Q module subunit and assembles early with the Q/P$_p$ subcomplex, while NDUFA6 assembles, among a handful of other accessory subunits, during the last step of N module attachment[19]. Therefore, NDUEG3 could only assemble after NDUFA6, otherwise coexisting of NDUEG3 and the early participant NDUFA9 would sterically preclude attachment of NDUFA6 (Supplementary Fig. 21b). This indicates that plugging of the

NDUEG3-NDUEG5 heterodimer onto Eg-CI happens very late during assembly, after the attachments of N module and final accessory subunits including NDUFA6 (Supplementary Fig. 21b). This hypothesis is in line with how the FAS domain is associated onto Eg-CI, as late assembly ensures that both its scaffold subunits NDUFA6 and NDUFS4 are already in place. Whether other FAS subunits pre-assemble with NDUEG3-NDUEG5 or are incorporated at alternative stages cannot be easily inferred (Supplementary Fig. 21b). The potential NDUFA6 requirement for the FAS domain association indicates that its function in the euglenozoan common ancestor could relate to the cellular fatty acid level reflected by the acylated ACP/LYR pair[19].

## Methods

### Cell culture and mitochondrial isolation—Euglena gracilis

Was obtained from Guangyu Biological Technology Co., LTD, Shanghai, China under the catalog number GY-D32 and cultured in liquid mineral Tris-minimum-phosphate medium (TMP) pH 7.0 supplemented by a pre-mixed vitamin solution ($10^{-7}$% biotin, $10^{-7}$% B$_{12}$ vitamin and $2 \times 10^{-5}$% B$_1$ vitamin (w/v)) and with 1% ethanol (v/v) as carbon source[74,75]. The following steps were performed at 4 °C unless otherwise stated. The cells were grown at 25°C with 120 rpm shaking in the dark and collected in the middle of the logarithmic phase by centrifugation at $1200 \times g$, 10 min. The cell pellet was immediately weighed and resuspended with EMIBS buffer (50 mM HEPES pH 7.5, 2 mM EDTA, 280 mM sucrose, 1 mM dithiothreitol (DTT), 0.002% phenylmethylsulfonyl fluoride (PMSF) (w/v)) at 3 ml/g cell mass. The cell resuspension was sonicated on ice using a Scientz-IID sonicator (SCIENTZ), with a microprobe of 15 mm tip diameter at 300 Watts output for 20 s followed by 20 s resting period, repeated twice. The sonicated sample was centrifuged at $600 \times g$ for 10 min, the supernatant of which was centrifuged again at $20,000 \times g$ for 20 min to pellet down the mitochondria. The crude mitochondria fraction was loaded onto a discontinuous sucrose gradient with 30%, 45% and 60% sucrose (w/v) in EMIB buffer (EMIBS without sucrose) by ultracentrifugation at $115,400 \times g$ for 2 h using an Optima XPN Ultracentrifuge and a SW32 Ti rotor (Beckman Coulter). Purified mitochondria fraction was collected from the 45–60% sucrose interface, diluted to a lower sucrose concentration by drop-wise addition of equal volume of EMIB buffer, then pelleted at $16,000 \times g$ for 45 min. The fine mitochondria fraction was weighed and stored at −80 °C until use.

### Purification of electron transport chain complexes and supercomplexes

The following steps were performed at 4 °C unless otherwise stated. To isolate the membrane fraction of *E. gracilis* mitochondria, the fine mitochondria pellet was thawed and homogenized in milli-Q water at 10 ml/g mitochondria mass using a KIMBLE Dounce tissue grinder. The homogenate was added with 150 mM final concentration KCl and homogenized again to release the peripheral membrane proteins, before centrifugation at $32,000 \times g$ for 45 min. The resulting mitochondrial membrane pellet was resuspended in buffer M10 (20 mM Tris pH 7.4, 50 mM NaCl, 1 mM EDTA, 2 mM DTT, 0.002% PMSF (w/v), 10% glycerol (v/v)) at 18 ml/g mitochondria mass by homogenization as a washing step, before centrifugation again at $32,000 \times g$ for 45 min. The resulting mitochondrial membrane pellet was weighed and resuspended in buffer M10 at 1.5 ml/g membrane mass by homogenization. The protein concentration was determined using a bicinchoninic acid (BCA) assay kit (Takara Bio Inc.) following the manufacturer's instructions. The protein sample was then diluted by buffer M90 (20 mM Tris pH 7.4, 50 mM NaCl, 1 mM EDTA, 2 mM DTT, 0.002% PMSF (w/v), 90% glycerol (v/v)) to a final glycerol concentration of 30% (v/v) and stored at −80 °C until use.

For complex I (CI) purification, mitochondrial membrane fraction containing ~20 mg total protein was thawed and solubilized in

buffer M10 containing 1% lauryl maltose neopentyl glycol (LMNG) (w/ v) added at a detergent-to-protein ratio of 4:1 (w/w) for 1 h with gentle agitation. The LMNG extracted sample was centrifuged at $16,000 \times g$ for 45 min, the supernatant of which was filtered with a 0.45-μm filter and loaded onto a 5-ml Q-Sepharose HP column (Cytiva) equilibrated in buffer Q-A (30 mM Tris, pH 7.4, 50 mM NaCl, 2 mM MgCl$_2$, 0.002% PMSF (w/v), 10% glycerol (v/v), 0.001% cholesteryl hemisuccinate (CHS) (w/v), 0.007% LMNG (w/v)) for anion exchange chromatography (ANX). The column was washed with 25 ml of buffer Q-A, then eluted by a 100 ml linear gradient with 0 to 70% buffer Q-B (buffer Q-A with 1 M NaCl). Fractions were checked by Tris-glycine blue-native polyacrylamide gel electrophoresis (BN-PAGE) and in-gel nitrotetrazoleum blue (NTB) assay using the NTB buffer (20 mM Tris pH 7.4, 150 mM NADH, 1.5 mg/ml NTB). Fractions with CI activity were pooled and concentrated to ~500 μl using a 100 kDa molecular weight cutoff (MWCO) centrifugal concentrator. The sample was then added with a 1:4 mixture of cardiolipin and dioleoyl phosphatidylcholine (DOPC) at 0.5 mg/ml final concentration, and immediately loaded onto a Superose 6 Increase 10/300 GL column (Cytiva) for size-exclusion chromatography (SEC) in SEC-Q buffer (30 mM Tris pH 7.4, 200 mM NaCl, 1 mM EDTA, 0.002% PMSF (w/v), 0.001% CHS (w/v), 0.007% LMNG (w/v)). Fractions containing Eg-CI eluted at ~12 ml and were concentrated to ~3.5 mg/ml using a 100 kDa MWCO centrifugal concentrator for immediate cryo-EM grid preparation. Deactivated Eg-CI was prepared likewise, except that the concentrated sample from ANX was incubated at 37 °C for 20 min before SEC.

For supercomplex purification, mitochondrial membrane fraction containing ~20 mg total protein was thawed and solubilized in buffer MX (30 mM HEPES pH 7.7, 150 mM potassium acetate, 0.002% PMSF, 10% (v/v) glycerol) with 2% digitonin at a detergent-to-protein ratio of 4:1 (w/w) for 1 h with gentle agitation. After centrifugation at $16,000 \times g$ for 45 min, the supernatant was loaded onto a continuous 20–50% (w/v) sucrose gradient in SEC-S buffer (30 mM Tris pH 7.4, 100 mM NaCl, 1 mM EDTA, 0.002% PMSF (w/v), 0.001% CHS (w/v), 0.1% glycodiosgenin (GDN) (w/v)) and ultracentrifuged at $140,000 \times g$ for 20 h using an Optima XPN Ultracentrifuge and a SW32 Ti rotor (Beckman Coulter). The gradients were then fractionated manually and assayed for CI activity as described above. Fractions containing Eg-SC I + III$_2$ + IV and Eg-SC III$_2$ + IV$_2$, as judged from in-gel activity assay, were pooled, concentrated and loaded onto the Superose 6 Increase 10/300 GL column (Cytiva) in SEC-S buffer for SEC. Fractions containing Eg-SC I + III$_2$ + IV and Eg-SC III$_2$ + IV$_2$ eluted at ~10 ml with a peak concentration of ~0.3 mg/ml. This peak fraction was used directly for cryo-EM grid preparation.

### Expression and purification of E. gracilis cytochrome c

Open reading frame (ORF) of the *E. gracilis* cyt *c* were obtained from the Uniprot database with accession number P00076. pET22b(+) plasmid containing the ORF sequences immediately between the pelB periplasmic signal sequence and the 6×His tag was synthesized and transformed into *E. coli* BL21 strain (Tsingke Biotechnology). Plasmid pEC86 containing the *E. coli* ccmABCDEFGH operon was purchased from the Swiss Culture Collection Center and transformed into the *E. coli* BL21 strain with the pET22b(+) plasmid for the maturation of cyt *c*[76]. A single double-transformed colony was cultured in 15 ml Lysogeny broth (LB) with 100 μg/ml ampicillin and 34 μg/ml chloramphenicol at 37 °C with 250 rpm shaking for about 12 h. 10 ml of this culture was used to inoculate 100 ml LB containing the same antibiotics and cultured identically for another 12 h. Next, 4 L LB with the same antibiotics were evenly distributed to eight 1 L conical flasks, each inoculated with 10 ml of previous culture and cultured identically for about 4 h until the optical density at 600 nm reached between 0.9 to 1. The cultures were then induced by 30 μM isopropyl β-D-1-thiogalactopyranoside (IPTG) and cultured at 30˚C with 190 rpm shaking overnight.

The following steps were performed at 4 °C unless otherwise stated. The bacterial cells were pelleted by centrifugation at $3000 \times g$ for 15 min. The cell pellets were resuspended in 120 ml ice-cold TES buffer (100 mM Tris pH 7.8, 20% sucrose w/v, 0.5 mM EDTA and 0.002% PMSF), added with 0.5 mg/ml lysozyme (Sangon Biotech) and incubated for 15 min at room temperature (RT). The resuspension was then added with 0.002% PMSF and bovine DNase I (Sigma-Aldrich) and homogenized by sonication on ice with a microprobe of 115 μm tip diameter at 300 Watts output for 3 s followed by 3 s resting period, repeated for eight times. The homogenate was centrifuged at $12,000 \times g$ for 30 min, the supernatant of which was filtered through a 0.45 μm filter and loaded onto a 5 ml SP Sepharose FF column (Cytiva) equilibrated in buffer S-A (100 mM Tris pH 7.8, 0.5 mM EDTA) for ANX. The column was washed with 25 ml buffer S-A and eluted with a 0–50% gradient of buffer S-B (buffer S-A with 1 M NaCl) over 20 column volumes. Peak fractions monitored by absorbance at 422 nm and 558 nm were pooled and concentrated with a 3 kDa MWCO centrifugal concentrator to final concentrations of 0.45 mM for Eg-cyt *c*, measured by BCA assay. The presence and size of the cyt *c* were confirmed by sodium dodecyl sulfate-polyacrylamide gel electrophoresis (SDS-PAGE) and coomassie brilliant blue staining. The cyt *c* stocks was stored at −80 °C until use.

To measure the extinction coefficient of the Eg-cyt *c*, fully oxidized and reduced Eg-cyt *c* were prepared by adding 0.03% H$_2$O$_2$ (v/v) or 2.5 mM DDT respectively. The redox states of above preparations were determined by a spectral scan from 200 nm to 800 nm with 1 nm increments using a Thermo Scientific Spectrophotometer, from which an A$_{558 nm}$/A$_{565 nm}$ ratio >9.0 was considered fully reduced. Standard curves for oxidized and reduced Eg-cyt *c* were plotted by measuring absorbance at 558 nm for 25 μM, 50 μM, 75 μM, 100 μM and 125 μM dilution of the 0.45 mM Eg-cyt *c* stock in three replicates. Values in the standard curves are expressed as averages ± standard error of mean (SEM). The reduced and oxidized extinction coefficients of Eg-cyt *c* were calculated by dividing the slopes of the two standard curves by the optical path length (0.2 cm), giving a reduced-oxidized Eg-cyt *c* extinction coefficient of 6.45 mM$^{-1}$ cm$^{-1}$.

### Spectroscopic assays for Eg-CI and Eg-SC I + III$_2$ + IV activity

Eg-ETC activities were determined spectroscopically at 340 nm and 558 nm as NADH oxidation and cyt *c* reduction and oxidation rates in the presence or absence of CI (rotenone, Sigma-Aldrich), CIII$_2$ (antimycin A, MKbio) and CIV (sodium azide, Sigma-Aldrich) inhibitors. All activities were measured as triplicates in 384-well plates at RT using a Thermo Scientific Spectrophotometer with a total reaction volume of 20 μl per well. Extinction coefficients of 6.22 mM$^{-1}$ cm$^{-1}$, 6.45 mM$^{-1}$ cm$^{-1}$ and 6.5 mM$^{-1}$ cm$^{-1}$ were used for reduced NADH, Eg-cyt *c* and horse cyt *c* in the calculations[6].

The reaction master mix consisted of 20 mM Tris-HCl pH 7.4, 50 mM NaCl, 0.1% digitonin (w/v), 0.002% PMSF and was added by protein samples, substrates and inhibitors wherever needed. For Eg-CI activity, 10 nM Eg-CI and 200 μM decylubiquinone (DQ) were used with or without the addition of 0–500 μM rotenone. The reaction was initiated by addition of 250 μM NADH, mixed in the spectrophotometer for 10 s and the 340 nm absorbance was recorded every 6 s for above 10 min. Eg-CI activity was calculated as NADH oxidation rate, based on the slope of the initial linear phase between 0–300 s in the 340 nm kinetic curve. For Eg-SC I + III$_2$ + IV activity, 10 nM Eg-SC I + III$_2$ + IV or *S. scrofa* SC I + III$_2$ + IV, 20 μM DQ, 100 μM Eg- or horse cyt *c* (Sigma-Aldrich) were used with or without relevant inhibitors (500 μM rotenone, 1 mM antimycin A or 100 mM sodium azide). The reaction was initiated by addition of 250 μM NADH, mixed in the spectrophotometer for 5 s, and then recorded every 6 s for ~20 min at 340 nm and 550 nm (horse cyt *c*) or 558 nm (Eg cyt *c*) simultaneously. NADH oxidation rate, cyt *c* reduction rate and cyt *c* oxidation rate were calculated based on the slopes of the initial linearly increasing phases

between 0–300 s in the 340 nm kinetic curves, the initial linearly increasing phases between 0 and 120 s in the 558/550 nm kinetic curves and the last linearly decreasing phases between 600 s and 900 s in the 558/550 nm kinetic curves.

## Spectroscopic assays for activities of the FAS domain components

For TER activity measurement, the reaction mixture consisted of 50 mM potassium phosphate pH 7.5, 0.1 mg/ml bovine serum albumin (BSA), 0.1% GDN, 2 µM FAD, 5 mM $MnCl_2$, 5 mM $MgCl_2$, 200 µM crotonyl-CoA or 2-trans-dodecenoyl-CoA, 20 nM SC I + III$_2$ + IV or 50 µg/ml *E. gracilis* isolated mitochondrial membranes or 50 µg/ml *Homo sapiens* 2-enoyl thioester reductase 1 (ETR1) (Solarbio Life Science). The reaction was started by the addition of 250 µM NAD(P)H, mixed in the spectrophotometer for 10 s and the 340 nm absorbance was recorded every 6 s for above 10 min. The TER activity was calculated as NAD(P)H oxidation rate based on the slope of the initial linear phase between 0–300 s in the 340 nm kinetic curve. For PEPCK activity measurement, the reaction mixture contained 30 mM Tris pH 7.4, 100 mM NaCl, 100 mM $NaHCO_3$, 5 mM $MnCl_2$, 5 mM $MgCl_2$, 500 µM NADH, 10 mM ADP or GDP, 1.2 U malate dehydrogenase (Sigma-Aldrich), 20 nM Eg-SC I + III$_2$ + IV or 50 µg/ml *E. gracilis* isolated mitochondrial membranes or 50 µg/ml *Zea mays* PEPCK (Shanghai Yuanye Bio-Technology). The reaction was started with the addition of 10 mM PEP, mixed in the spectrophotometer for 10 s and the 340 nm absorbance was recorded every 6 s for above 10 min. The PEPCK activity was calculated as NADH oxidation rate based on the slope of the initial linear phase between 0–300 s in the 340 nm kinetic curve.

## Spectroscopic assays for CI's active-to-deactive (A/D) transition

To measure CI activity, the reaction master mix consisted of 20 mM HEPES pH 7.4, 50 mM NaCl, 10% glycerol (v/v), 0.1% BSA (w/v), 0.1% 3-[(3-cholamidopropyl)dimethylammonio]−1-propanesulfonate (CHAPS) (w/v), 0.1% GDN (w/v) and 200 µM DQ. For the thermal deactivation of CI and SC I + III$_2$ + IV, 10 nM protein sample was added to the reaction mix and the plate was incubated at 37 °C for 20 min, after which 10 µM NADH or equivalent amount of buffer was added to the well, mixed by pipetting and incubated in the spectrophotometer for 30 s. Afterwards, 2.5 mM or 5 mM $MgCl_2$, or 0.5 mM or 2 mM N-ethylmaleimide (NEM), or equivalent amount of buffer was added to the corresponding wells and mixed in the spectrophotometer for 2 min. For the $MgCl_2$ assay, the reactions were initiated by the addition of 300 µM NADH and immediately mixed by the spectrophotometer for 10 s before recording the 340 nm absorbance every 6 s for above 10 min. The NEM assay used a similar set-up, except that the plate was incubated for 15 min in the dark at RT after addition of the NEM. Eg-CI activity was calculated as NADH oxidation rate, based on the slope of the initial linear phase between 0–200 s in the 340 nm kinetic curve.

## Cryo-EM grid preparation and data collection

The cryo-EM grid preparation and data collection were carried out at the Center of Cryo-Electron Microscopy at Zhejiang University. For Eg-CI deactivated dataset (Dataset 2), thermally deactivated Eg-CI sample was used for vitrification without additional treatment. For Eg-CI NADH reduced dataset (Dataset 4), 9 µl native Eg-CI sample was incubated with 1 µl 50 mM NADH for 2 min at 4 °C immediately before vitrification. For Eg-CI turnover dataset (Dataset 3), 8 µl native Eg-CI sample was incubated with 1 µl 10 mM DQ and 2% CHAPS for 20 s at 4 °C, then 1 µl of 50 mM NADH was added to the sample immediately before vitrification. For each of the three Eg-CI datasets, 3 µl sample was applied to pre-glow discharged Quantifoil 0.6/1 copper grids and blotted by Vitrobot Mark IV (Thermo Fisher) for 3 s under 100% humidity at 4 °C before being plunged into liquid ethane. Grids were transferred to liquid nitrogen for storage before data collection. Cryo-EM grid preparation for Eg-SC I + III$_2$ + IV and SC III$_2$ + IV$_2$ (Dataset 1)

was similar, except that a Quantifoil R1.2/1.3 300 mesh copper grid with 2 nm continuous layer of carbon was used.

Micrographs were acquired on a Titan Krios microscope (Thermo Fisher) operating at 300 kV equipped with a Falcon4 detector operating at 241 frames/s and a Selectris energy filter. EPU software was used for automated data collection following standard procedures. For Datasets 2–4, the grids were imaged at a calibrated magnification of ×130,000 with a physical pixel size of 0.93 Å and a defocus range from −0.8 to −2.0 µm. A total dose of 61.5 e⁻/Å² with 7 s exposure time was fractionated into 1687 frames. A total of 2702, 6758 and 2949 movies were collected for Datasets 2–4 respectively. For Dataset 1, the grid was imaged at a calibrated magnification of ×105,000 with a physical pixel size of 1.2 Å and a defocus range from −0.6 to −1.8 µm. A total dose of 51.51 e⁻/Å² with 7.82 s exposure time was fractionated into 1884 frames and a total of 9807 movies were collected.

## Cryo-EM image processing

For all collected datasets, motion-correction was performed using Relion-4.0's own implementation[77], before contrast transfer function (ctf) estimation using CTFFIND4.1[78]. Particles were picked in crYOLO[79] and extracted in Relion-4.0 using box sizes 448 for Dataset 1 and 480 for Datasets 2–4. The following steps were performed in cryoSPARC v3.3.2[80]. For Datasets 2–4, a total of 66,800, 86,599 and 76,643 Eg-CI particles were generated after particle curation by 2D classification and 3D ab-initio reconstruction. Overall non-uniform refinement[81] with global and local ctf corrections and C1 symmetry gave resolutions of 3.11 Å, 3.03 Å and 2.87 Å, respectively. For each dataset, local refinements were performed on Eg-CI PA, Eg-CI MA-proximal and Eg-CI-MA distal regions. Composite maps of these regional maps form each dataset were then generated using the Combine Focused Maps program in Phenix-1.20.1[82].

For Dataset 1, a total of 1,374,054 particles with defined structural features were obtained after particle curation by 2D classification. Heterogeneous refinement separated these particles into four classes corresponding to Eg-SC I + III$_2$ + IV, Eg-SC III$_2$ + IV$_2$, Eg-SC V$_2$ and Eg-SC V$_4$, with 345,048, 135,598, 263,948 and 239,781 particles respectively. Overall non-uniform refinement with lobal and local ctf corrections and C1 symmetry gave resolutions of 2.81 Å and 3.06 Å for Eg-SC I + III$_2$ + IV and Eg-SC III$_2$ + IV$_2$ respectively. For Eg-SC I + III$_2$ + IV particles, local refinements were performed on Eg-CI PA, Eg-CI MA-proximal region, Eg-CI-MA distal region, Eg-CIII$_2$ and Eg-CIV, giving resolutions of 2.73 Å, 2.72 Å, 2.69 Å, 2.79 Å and 2.76 Å respectively. For Eg-SC III$_2$ + IV$_2$ particles, local refinements were performed on Eg-CIII$_2$ and the two Eg-CIVs, giving resolutions of 2.85 Å, 2.89 Å and 3.14 Å respectively. Composite maps of these regional maps from both particle sets were generated using the Combine Focused Maps program in Phenix-1.20.1[82], which were used for the following manual model building and automatic model refinements.

Focused 3D classifications without alignment were perform in Relion-4.0[77] on the CI PA and CIV regions of Eg-SC I + III$_2$ + IV using locally refined particles focusing on the CI MA proximal and CI MA distal regions respectively. 3D classification on CI PA gave two sensible classes with 167,886 and 166,802 particles, which we named state 1 and state 2 respectively. However, the MA-PA angle did not have obviously difference between these two states, nor did the conformations of the Q tunnel loops and ND6 TMH3. 3D classification on CIV gave four sensible classes with 31,333, 40,180, 228,091 and 34,785 particles with gradually increasing angle between CIV and CIII$_2$. We named these classes state close, state intermediate 1, 2 and state open respectively. Local refinements with full Eg-SC I + III$_2$ + IV mask were performed for these altogether six sensible classes. For Eg-SC III$_2$ + IV$_2$, focused 3D classifications without alignment were perform in Relion-4.0[77] separately on the left and right CIV regions using locally refined particles focusing on the CIII$_2$. Four and three sensible classes were obtained from these two 3D classifications, which we named states closed-L,

intermediate 1 L, intermediate 2 L and state open-L for classification on the left CIV, as well as states closed-R, intermediate-R and open-R for classification on the right CIV. Local refinements with full Eg-SC I + III$_2$ + IV mask were performed for these altogether seven sensible classes. Focused 3D variability analysis (3DVA) of the CIII$_2$ Rieske head domains was performed using masks around the IMS parts of CIII$_2$ in both Eg-SC I + III$_2$ + IV and Eg-SC III$_2$ + IV$_2$ particle sets. Both refinements were low-pass filtered to 5 Å resolution that all other parameters were kept as default in cryoSPARC using their built-in algorithm[83].

## Model building and refinement

All manual model building was performed in Coot-0.9.6[84]. Mainchain backbones of core and conserved subunits were built by docking plant ETC models (PDB 7AR8 [https://www.rcsb.org/structure/7AR8] (*A. thaliana* CI), 7ARD [https://www.rcsb.org/structure/7ARD] (*Polytomella* CI), 7JRG [https://www.rcsb.org/structure/7JRG] (*V. radiata* CIII$_2$) and 7JRO [https://www.rcsb.org/structure/7JRO] (*V. radiata* CIV)) into respective local refinement maps of Eg-SC I + III$_2$ + IV by rigid-body fitting in ChimeraX[11,85,86]. Mainchain backbones of Eg-specific subunits were built de novo according to density. Side chains of above built backbones were added de novo according to density and the generated query sequences were used in BLAST searches against the *E. gracilis* genome and transcriptome (accession codes: PRJEB27422 and GCA_900893395.1). Assembly information of certain transcripts were based on previous electrophoretic and mass spectroscopic studies of the *E. gracilis* ETC[28,87]. Secondary structure information predicted by the PredictProtein server was also used to assist modeling in Coot-0.9.6[88]. The manually built Eg-CI, Eg-CIII$_2$ and Eg-CIV models were rigid-body fitted into the composite Eg-SC I + III$_2$ + IV and Eg-SC III$_2$ + IV$_2$ maps with chain ID adjusted according to the previously published ciliate ETC structures[7]. Automatic model refinements were performed using the phenix.refine and phenix.real_space_refine programs against the composite Eg-SC I + III$_2$ + IV and Eg-SC III$_2$ + IV$_2$ maps. We automatically generated secondary structure restraints, custom bond linkage and custom ligand description file by Phenix-1.20.1 before manually editing according to the outcome of the automatic refinement. The refined model was manually edited in Coot-0.9.6 before the next round of automatic refinement until the refined model achieved high model-map correlation and good geometric statistics. A total of 60, 22 and 19 subunits were built for Eg-CI, Eg-CIII$_2$ and Eg-CIV, among which 12, 1 and 9 subunits were Eg-specific respectively. The Eg-CI part of the refined Eg-SC I + III$_2$ + IV model was copied out and rigid-body fitted into the composite Eg-CI maps under deactivated, NADH reduced and turnover conditions. To generate initial Eg-CI models for Datasets 2–4, residues were deleted where clear density could not be found in respective composite maps. The initial models were first automatically refined against respective composite maps using the phenix.refine and phenix.real_space_refine programs, outcomes of which were then manually edited for the next round of automatic refinement until the refined model achieved high model-map correlation and good geometric statistics.

## Molecular dynamics simulation

The structure of *E. gracilis* cytochrome c (Eg-cyt c) was predicted using AlphaFold2 (AF2)[89] installed on our local computer cluster with the default pipeline. An initial model of the Eg-CIV in complex with Eg-cyt c was prepared by aligning the AF2-predicted Eg-cyt c with the equine cytochrome c (Ec-cyt c) and Eg-CIV with bovine CIV (Bt-CIV) from the crystal structure of the Bt-CIV+Ec-cyt c complex (PDB 5IY5 (crystal structure of mammalian CIV in complex with cyt c)). To reduce computational cost, the Eg-specific matrix domain was removed, retaining only the TM and IMS domain of Eg-CIV. Metal ions were kept and heme a and a$_3$ molecules were replaced by heme c. The model did not contain the resolved lipids but was embedded into a flat, mixed lipid bilayer composed of 50% 1-palmitoyl-2-oleoyl-phosphatidylcholine

(POPC), 30% 1-palmitoyl-2-oleoyl-phosphoethanolamine (POPE), and 20% 1′,3′-bis[1-palmitoyl-2-oleoyl-sn-glycero-3-phospho]-glycerol (16:0–18:1 cardiolipin), and solvated in a cubic water box containing 0.15 M NaCl. The box dimension was 13.0 nm × 13.0 nm × 16.0 nm, resulting in approximately 276,000 atoms for the Eg-CIV+Eg-cyt c model. The OPM webserver[90] was used to orient the transmembrane domain of Eg-CIV within the lipid bilayer, and the system was constructed using the CHARMM-GUI webserver[91]. Following energy minimization with the steepest descent algorithm and equilibration in six steps by gradually removing position constraints, production runs were conducted under semi-isothermal-isobaric (NPT) conditions using the CHARMM36m force field[92] for proteins, CHARMM36 for lipids, TIP3P for water, and CGenFF for heme c[93]. The two copper ions in the dinuclear Cu$_A$ were connected with a bond potential. Distance restraints were introduced between the iron atom of heme c and the adjacent histidine and methionine residues in Eg-cyt c using the PLUMED plugin[94]. The temperature was maintained at 310 K using a Nose-Hoover thermostat and the pressure at 1.0 bar using the Parrinello-Rahman barostat. A cut-off of 1.2 nm was applied for van der Waals and short-range electrostatic interactions, with a switch function at 1.0 nm. Long-range electrostatic interactions were calculated using the particle mesh Ewald algorithm, featuring a mesh spacing of 0.12 nm. Four independent simulations were performed for the atomistic model, with each initially lasting 1000 ns. To access the stability of Eg-cyt c binding to Eg-CIV, three simulations demonstrating satisfactory cyt c binding were further extended to 2000 ns, accumulating in a total simulation time of 7000 ns.

To investigate the membrane reorganization surrounding the complex, coarse-grained (CG) molecular dynamics simulations were performed on the Eg supercomplex structure utilizing the Martini3 force field[95]. The atomistic structure was converted into a CG model using the martinize2 tool (with vermouth 0.7.3). Cofactors and resolved lipids from the cryo-EM structures were removed, and only protein domains associated with membrane binding were simulated to minimize computational expenses. The insane.py script[96] was used to embed the CG supercomplex in a large lipid bilayer with a POPC:POPE ratio of 7:3. The elastic-network approach was applied to constrain the protein structure, using a force constant of 1000 kJ/mol/nm$^2$ and cutoff distance limits of 0.5 and 0.9 nm. For comparison, the *T. thermophila* SC I + III$_2$ was also modeled using its cryo-EM structure (PDB 7TGH (*T. thermophila* SC I + III$_2$)) as a template. The systems were solvated with a CG Martini water model and neutralized by adding NaCl at a concentration of 0.15 M. The box sizes for the Eg and Tt models were 50 × 50 × 25 nm$^3$ and 50 × 50 × 20 nm$^3$, respectively. Initially, the CG system was minimized for 5000 steps using the steepest descent method, followed by equilibration according to the standard six-step CHARMMGUI equilibrium protocol. Each production run was carried out in the semi-isotropic NPT ensemble with a 20 fs time step. During production runs, 500 kJ mol$^{-1}$ nm$^{-2}$ harmonic constraints were maintained on the backbone (Martini BB beads). The system's temperature was held at 310 K using the velocity rescaling thermostat, while the pressure was maintained at 1 bar with the Parrinello-Rahman barostat, a compressibility of $3 × 10^{-4}$ bar$^{-1}$, and a coupling constant of 12 ps. Each production run was performed for 500 ns and the last frame of the MD trajectory was used to represent the membrane curvature.

Gromacs 2022.5[97] with GPU acceleration was employed for all simulations. Trajectory analysis was performed using Gromacs gmx tools, PLUMED driver and GetContacts (https://getcontacts.github.io/). The reliability and reproducibility of our MD simulations are summarized as Supplementary Table 9.

## Reporting summary

Further information on research design is available in the Nature Portfolio Reporting Summary linked to this article.

## Data availability

The composite maps and models for Eg-SC I + III$_2$ + IV, Eg-SC III$_2$ + IV$_2$ and Eg-CI are available in the Electron Microscopy Database (EMDB) and the Protein Data Bank (PDB) with accession codes as following: Eg-SC I + III$_2$ + IV EMDB-35720, PDB-8IUF; Eg-SC III$_2$ + IV$_2$ EMDB-35723, PDB-8IUJ; Eg-CI turnover EMDB-36108, PDB-8J9I; Eg-CI NADH reduced EMDB-36109, PDB-8J9J; Eg-CI deactivated EMDB-36107, PDB-8J9H. Consensus and local refinement maps for Eg-SC I + III$_2$ + IV, Eg-SC III$_2$ + IV$_2$ and Eg-CI are available in the EMDB with accession codes as following: Eg-SC I + III$_2$ + IV consensus map EMDB-35819, CI-PA region EMDB-35662, CI-MA proximal region EMDB-35663, CI-MA distal region EMDB-35664, CIII$_2$ region EMDB-35665 and CIV region EMDB-35666; Eg-SC III$_2$ + IV$_2$ consensus map EMDB-35820, CIII$_2$ region EMDB-35667, two CIV regions EMDB-35668 and EMDB-35669; Eg-CI turnover state consensus map EMDB-36102, CI-PA region EMDB-36099, CI-MA proximal region EMDB-36100, CI-MA distal region EMDB-36101; Eg-CI NADH reduced state consensus map EMDB-36106, CI-PA region EMDB-36103, CI-MA proximal region EMDB-36104, CI-MA distal region EMDB-36105; Eg-CI NADH deactivated state consensus map EMDB-36098, CI-PA region EMDB-36094, CI-MA proximal region EMDB-36096, CI-MA distal region EMDB-36097. For model building, *Euglena gracilis* genome and transcriptome used are obtained from GenBank under accession code GCA_900893395.1. Structures used for initial model building are available in the PDB under accession codes 7AR8, 7ARD, 7JRG and 7JRO. Structures used for comparison and illustration purposes are available in the PDB under accession codes 1AQ2, 1GUF, 1HET, 1HRC, 1II2, 2FUG, 2VCY, 3EBQ, 3R44, 4WAS, 5IY5, 5J4Z, 5Z62, 6QBX, 6RFR, 6T0B, 6TDU, 6ZKD, 6ZKH, 6ZKK, 6ZKO, 6ZKP, 6ZKS, 7AR8, 7ARD, 7JRG, 7JRO, 7JRP, 7O37, 7O3C, 7QSK, 7TGH, 8BPX, 8E73. For MD simulations and data presentation, *Euglena gracilis* cytochrome *c* and mitochondrial TER sequences are obtained from Uniprot under accession codes P00076 and Q5EU90, respectively. Raw data for gel images and activity measurements are provided in the Source Data file. Source data are provided with this paper.

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

## Acknowledgements

We thank Shenghai Chang and Lingyun Wu in the Center of Cryo-Electron Microscopy (CCEM), Zhejiang University for their technical assistances on Cryo-EM grid screenings and data collections. We thank Cheng Ma and Liyan Wang from the Core facilities, Zhejiang University School of Medicine for their supports with protein purification and characterization. We acknowledge the computational support of the Information Technology Center and State Key Lab of CAD&CG at Zhejiang University. This project is supported by the ZJU100 Young Professor grant from School of Medicine, Zhejiang University (to L.Z. and Y.W.), the National Natural Science Foundation of China (32371253 to L.Z., and 32371300 to Y.W.), National Key Research and Development Program of China (No. 2021YFF 1200404 to Y. W.), the Zhejiang Provincial National Science Foundation of China (No. LZ24C050003 to Y. W.), and Major Science and Technology Project of Zhejiang Province Health Commission (No.WKJ-ZJ-2112 to J.C.Z.).

## Author contributions

Conceptualization, L.Z.; Methodology, L.Z., Y.W. and J.C.Z.; Investigation, Z.X.H., M.C.W., H.T.T., L.D.W., Y.Q.H., F.Z.H., J.C.Z., Y.W. and L.Z.; Writing—Original Draft, L.Z., Z.X.H.; Writing—Review & Editing, L.Z., Y.W. and J.C.Z.; Visualization, H.T.T., Z.X.H., M.C.W., Y.W. and L.Z.; Funding Acquisition, L.Z., Y.W. and J.C.Z.; Resources, L.Z., Y.W. and J.C.Z.; Supervision, L.Z. and Y.W.; Project administration, L.Z.

## Competing interests

The authors declare no competing interests.
