## [Peer Review File · Nature Communications]

Euglena's Atypical Respiratory Chain Adapts to the Discoidal Cristae and Flexible MetabolismEditorial Note: This manuscript has been previously reviewed at another journal that is not operating a transparent peer review scheme. This document only contains reviewer comments and rebuttal letters for versions considered at *Nature Communications*.

REVIEWER COMMENTS

Reviewer #1 (Remarks to the Author):

In this revised manuscript by He et al, entitled “Euglena’s Atypical Respiratory Chain Adapts to the Discoidal Cristae and Flexible Metabolism” the authors have addressed the multiple concerns raised in the initial revision. The main text is better understandable and more aligned with the figures in the revised version. I am satisfied with most of the responses the authors provided to the concerns raised in the previous revision. However, I feel that representation of data is still confusing in the main text and a better representation will be helpful in critical evaluation of the data by the journal readership.

1. The authors changed the color scheme in the main text figures in revised version but a better representation especially indicating which one is CI, CIII and CIV in the Eg-SC super complexes figures will be more helpful. Also, in the figure panels where the super complexes are compared are confusing, a side by side comparison may be helpful to better understand the uniqueness of the Eg-SC’s compared to the previously known SC’s. Since a lot of subunits are discussed, careful representation of their names in the text, figures and figure legends is needed.

Examples:

-In figure 3c, *T. thermophila* (PDB 7TGH) is not shown but indicated in the figure legend. The comparison of the angles is also difficult due to confusing superimposition.

-In figure Supplementary 13e, the subunit name is UQCR10 while its name in the text is UQCREG1, I assume it’s a typo.

2. Location of FAS domain on C1 is one of the major highlights of the present work but its representation especially in Figure 2 is vague as subunits of this domain are not clearly defined.

3. The split core subunits are another highlight of this work so these subunits can be better shown as zoomed in insets of the main text figures. For example an inset figure describing the statement “In Eg-CI, the Eg-specific NT loop of NDUFS1A covers this groove and turns it into a ‘eyelet’ where NDUFS4’s CT loop can only thread through NDUFS1A and NDUFS1B” will be helpful. Also, the authors have mentioned

that the electron density for split domains is well defined, a supplementary figure with electron density will be helpful for the reader.

4. Detailed description of each figure panel at least of the main text will be very helpful for the readers.

5. Figure 4f,g the comparison of local conformations of the Q tunnel loops is hard to visualize in the figure panel.

Reviewer #2 (Remarks to the Author):

The authors have answered in an appropriate manner to my comments. I appreciate the modified discussion and the supplemental figures added.

I have no further comments.

Reviewer #3 (Remarks to the Author):

Contributions:

Authors report the high resolution structures of Euglena SCs I+III2+IV and III2+IV2. These structures clarify details about particular features of the Euglena ETC, such as fatty acid synthesis (FAS) and helmet-like domains in complexes I and IV, respectively. Using molecular dynamics simulations (MD), the authors propose a unique supercomplex organization that adapt to the negative membrane curvature which could contribute to fold the discoidal cristae described in this lineage. Also, they cover the active-deactive transition of Euglena CI and the crucial orientation of CIII2 and CIV in their SCs. Finally, based also on MD predictions, authors hypothesize about the binding mechanism of Euglena cyt c to CIV.

The work contributes to expand the knowledge about the structural divergences on ETC complexes among eukaryotic lineages and their consequences on stability and function. I found quite interesting the presented findings, the contribution to the field is clear, expanding the studied lineages, nevertheless some of the conclusions should be clarified or better supported.

Major points

Main text

- A deeper management of the eukaryotic evolution should be reflected as a pillar of the scope, without this, the manuscript focuses only on a detail structural description and gives the impression of been just the report of an “atypical” respirasome structure.

- Lines 26-28. A structural FAS domain has been previously identified on isolated *Euglena* CI (Miranda-Astudillo et al., 2018), additionally, based on complete cell studies the process of anaerobic wax fermentation has been proposed to be linked to CI, electron transfer flavoprotein (ETF), and rhodoquinone (RQ) in *Euglena* mitochondria (Nakazawa et al., 2018). Nevertheless, no evidences of the catalytic activity of the subunits involved in FAS domain are shown in the present study. Without these evidences, the affirmation put forward on the abstract: “A unique fatty acid synthesis domain locates on the peripheral arm tip of its complex I, linking it to the process of anaerobic wax fermentation” is misleading.

- Lines 253-256. Authors remark a deactivation of *Euglena* CI and respirasome in presence of N-ethylmaleimide (NEM) similar than the one observed on *Sus scrofa* CI and respirasome but which is not changed upon NADH re-activation. Even though this claim is partially true, the more notable effect, at least comparing CI graphs, is that the deactivation of *Euglena* CI is a lot less compared with mammalian CI, with this in mind, there is no evident “re-activation” because there is no remarkable “deactivation” to begin with. Authors should rephrase their conclusions.

- The discussion section on main text is only focused on the CI FAS domain biogenesis and the putative explanation of its acquisition on *Euglena* CI. No comments about the role of the helmet domain on CIV are signaled within the main text, some lines should be included about this unique structural feature, whose name comes from a previous work (Miranda-Astudillo et al., 2018), authors should pay attention specially on the cyt c – CIV stable interaction which allowed in the past the demonstration of the full in vitro NADH-O₂ electron transport showed previously on the purified respirasome from this species (Miranda-Astudillo et al., 2021). This is important because previous attempts to measure the in vitro oxidase activity using exogenous cyt c as electron donor revealed a specific requirement of *Euglena* complex IV for its endogenous cyt c (Brönstrup and Hachtel, 1989; Collins et al., 1975). Without these points included into the discussion, the section of the putative binding mechanism of *Euglena* cyt c to CIV based on molecular dynamics is feeble.

Supplemental material

- As expected, the molecular dynamics analysis (based on a *Eg*-cyt c alphafold model) performed twice to elucidate the cyt c – CIV binding site generated two different cyt c binding sites. From these results, authors selected the site which explains better their hypothesis, leaving the other one as a “trap site”,

where canonical cyt c could be trapped, explaining the previously observed impossibility of Euglena CIV to reduce exogenous cyt c (Brönstrup and Hachtel, 1989; Collins et al., 1975). In previous works, Euglena respirasome extracted from mitochondrial membranes using GDN detergent (the detergent used in the present manuscript to purify the respirasome by the sucrose gradient) could be purified with the endogenous cyt c attached as could be corroborated by spectroscopy and the measure of full NADH – O₂ electron flow (Miranda-Astudillo et al., 2021). These indicates a strong and stable binding property, which is not observed in the presented manuscript. Authors should take advantage on the technological tasks presented in their manuscript, and a reconstitution between the purified CIV (or any SC I/III2/IV or III2/IV2) and the already purified recombinant Eg-cyt c should be performed in a similar way as previously showed with the bovine oxidase and horse cyt c (Shimada et al., 2017) to elucidate the correct binding mechanism. Without these proper analysis, the proposed CIV-cyt c mechanism is speculative.

Minor points

Main text

- lines 40 and 41. “IV2+I+III2+II” and “IV2+(I+III2+II)” represent the same SC, check stoichiometry.
- line 51. Remove the term “protist”, nowadays this term is obsolete, and the recent evidence showed that the text-book kingdoms, e.g. animals, plants, fungi and protist, does not reflect the eukaryotic diversity and complexity (Burki et al., 2020; Keeling and Burki, 2019).
- lines 53 and 370. Recent studies addressed the split of the former Excavata supergroup into new supergroups, Euglenoids should be included into the Discoba supergroup (Burki et al., 2020).
- line 56. It should be important to clarify the secondary origin of the plastid in this species, because after each endosymbiotic process massive gene migrations were performed, contributing to the genetic wealth present in these domains (Leister, 2005).
- line 66. It reads: “to fill the missing gap...”, it should be read: “to fill a missing gap...”, even though this manuscript contributes to expand our knowledge among bioenergetics of eukaryotic organisms, there are still a lot of missing information related to other eukaryotic supergroups.

- line 191. A comparison of *Euglena* respirasome against alveolate (PDB 7TGH) respirasome is indicated, but in the corresponding figure 3(c), only *O. aries*, and *A. thaliana* are visible alongside the *Euglena* structure.
- line 715. Reference 55 listed on “Methods references” is not related to the heterotrophic culture of *Euglena gracilis*, and the conditions described in lines 713 - 715 resemble to previous reported conditions (Yadav et al., 2017). Please correct this reference.
- Figure 2 (g-i) For clarity, please indicate if there is any rotation of the “point of view” from the corresponding images.
- Figure 3 (c) *T. thermophila* (PDB 7TGH) SC is labeled in the figure legend, but is not visible in the corresponding panel.

References

- Brönstrup, U., Hachtel, W., 1989. Cytochrome c oxidase of *Euglena gracilis*: Purification, characterization, and identification of mitochondrially synthesized subunits. *J. Bioenerg. Biomembr.* 21, 359–373. <https://doi.org/10.1007/BF00762727>
- Burki, F., Roger, A.J., Brown, M.W., Simpson, A.G.B., 2020. The New Tree of Eukaryotes. *Trends Ecol. Evol.* 35, 43–55. <https://doi.org/10.1016/j.tree.2019.08.008>
- Collins, N., Brown, R.H., Merrett, M.J., 1975. Oxidative phosphorylation during glycollate metabolism in mitochondria from phototrophic *Euglena gracilis*. *Biochem J* 150, 373–377.
- Keeling, P.J., Burki, F., 2019. Progress towards the Tree of Eukaryotes. *Curr. Biol.* 29, R808–R817. <https://doi.org/10.1016/j.cub.2019.07.031>
- Leister, D., 2005. Origin, evolution and genetic effects of nuclear insertions of organelle DNA. *Trends Genet.* 21, 655–663. <https://doi.org/10.1016/j.tig.2005.09.004>

- Miranda-Astudillo, H., Yadav, K.N.S., Boekema, E.J., Cardol, P., 2021. Supramolecular associations between atypical oxidative phosphorylation complexes of *Euglena gracilis*. *J. Bioenerg. Biomembr.* 53, 351–363. <https://doi.org/10.1007/s10863-021-09882-8>
- Miranda-Astudillo, H., Yadav, K.N.S., Colina-Tenorio, L., Bouillenne, F., Degand, H., Morsomme, P., Boekema, E.J., Cardol, P., 2018. The atypical subunit composition of respiratory complexes I and IV is associated with original extra structural domains in *Euglena gracilis*. *Sci. Rep.* 8, 9698. <https://doi.org/10.1038/s41598-018-28039-z>
- Nakazawa, M., Ando, H., Nishimoto, A., Ohta, T., Sakamoto, K., Ishikawa, T., Ueda, M., Sakamoto, T., Nakano, Y., Miyatake, K., Inui, H., 2018. Anaerobic respiration coupled with mitochondrial fatty acid synthesis in wax ester fermentation by *Euglena gracilis*. *FEBS Lett.* 592, 4020–4027. <https://doi.org/10.1002/1873-3468.13276>
- Shimada, S., Shinzawa-Itoh, K., Baba, J., Aoe, S., Shimada, A., Yamashita, E., Kang, J., Tateno, M., Yoshikawa, S., Tsukihara, T., 2017. Complex structure of cytochrome c–cytochrome c oxidase reveals a novel protein–protein interaction mode. *EMBO J.* 36, 291–300. <https://doi.org/10.15252/embj.201695021>
- Yadav, K.N.S., Miranda-Astudillo, H. V., Colina-Tenorio, L., Bouillenne, F., Degand, H., Morsomme, P., González-Halphen, D., Boekema, E.J., Cardol, P., 2017. Atypical composition and structure of the mitochondrial dimeric ATP synthase from *Euglena gracilis*. *Biochim. Biophys. Acta - Bioenerg.* 1858, 267–275. <https://doi.org/10.1016/j.bbabi.2017.01.007>

Reviewer #1 (Remarks to the Author):

In this revised manuscript by He et al, entitled “Euglena’s Atypical Respiratory Chain Adapts to the Discoidal Cristae and Flexible Metabolism” the authors have addressed the multiple concerns raised in the initial revision. The main text is better understandable and more aligned with the figures in the revised version. I am satisfied with most of the responses the authors provided to the concerns raised in the previous revision. However, I feel that representation of data is still confusing in the main text and a better representation will be helpful in critical evaluation of the data by the journal readership.

1. The authors changed the color scheme in the main text figures in revised version but a better representation especially indicating which one is CI, CIII and CIV in the Eg-SC super complexes figures will be more helpful. Also, in the figure panels where the super complexes are compared are confusing, a side by side comparison may be helpful to better understand the uniqueness of the Eg-SC’s compared to the previously known SC’s. Since a lot of subunits are discussed, careful representation of their names in the text, figures and figure legends is needed.

The authors thank the reviewer for pointing out the color code issue regarding individual complexes in the supercomplex. In Figs. 1-3, apart from places where colors are defined by individual subunits, CI, CIII₂ and CIV are consistently colored as blue, green and magenta. To illustrate this point, color labels are added to Figs. 1-3 and the legends are revised to clarify this point.

Regarding the superimposed respirasome issue, Fig. 3c has now been revised so that Eg- and ovine respirasomes are shown side by side. This representation however creates some difficulties in comparing respective CIII₂ and CIV positions, since now they are in separate panels. We introduced color coded lines to mark CIII₂ and CIV positions in the Eg- and ovine panels in Fig. 3c so that translational distances and rotational angles can be labeled.

Examples:

-In figure 3c, *T. thermophila* (PDB 7TGH) is not shown but indicated in the figure legend. The comparison of the angles is also difficult due to confusing superimposition.

The authors thank the reviewer for pointing out this inaccuracy. In Fig 3c, *E. gracilis* respirasome is compared to its mammalian and plant counterparts, while in Fig 3d, the comparison is done among *E. gracilis*, mammalian and *T. thermophila* respirasome/SC I+III₂. The legend for Fig. 3 is revised to clarify this point.

Regarding the superimposed respirasome issue, Fig. 3c has now been revised so that Eg- and ovine respirasomes are shown side by side. This representation however creates some difficulties in comparing respective CIII₂ and CIV positions, since now they are in separate panels. We introduced color coded lines to mark CIII₂ and CIV positions in the Eg- and ovine panels in Fig. 3c so that translational distances and rotational angles can be labeled.

-In figure Supplementary 13e, the subunit name is UQCR10 while its name in the text is UQCREG1, I assume it’s a typo.

The authors thank the reviewer for pointing out this typo. The figure label has been corrected.

2. Location of FAS domain on C1 is one of the major highlights of the present work but its representation especially in Figure 2 is vague as subunits of this domain are not clearly defined.

Fig. 2a has been enlarged to clearly indicate position of the FAS domain within the respirasome. Meanwhile Fig. 2b has been changed to zoom-ins of the FAS domain, with each subunit shown as cylindrical cartoon so that internal structural relationship among these subunits are visualized.

3. The split core subunits are another highlight of this work so these subunits can be better shown as zoomed in insets of the main text figures. For example an inset figure describing the statement “In Eg-CI, the Eg-specific NT loop of NDUFS1A covers this groove and turns it into a ‘eyelet’ where NDUFS4’s CT loop can only thread through NDUFS1A and NDUFS1B” will be helpful. Also, the authors have mentioned that the electron density for split domains is well defined, a supplementary figure with electron density will be helpful for the reader.

We acknowledge that ‘NDUFS4 threads through the eyelet formed by NDUFS1A and NDUFS1B’ is not adequately visualized in Fig. 2g. To correct this, we updated Fig. S15j which now clearly illustrates the ‘eyelet’ and how NDUFS4 goes through it. Figure citations in the main and supplementary texts are updated accordingly.

We also updated supplementary figure 10 to include a panel corresponding to densities of NDUFS1A CT and NDUFS1B NT to illustrate that splitting of CI core subunit NDUFS1 is genuine in *Euglena gracilis* electron transport chain.

4. Detailed description of each figure panel at least of the main text will be very helpful for the readers.

The authors thank the reviewer for pointing out issues regarding the figure legends. The main and supplementary figure legends have been updated to increase clarity. Clubbing of panels in the legends are avoided as much as possible.

5. Figure 4f,g the comparison of local conformations of the Q tunnel loops is hard to visualize in the figure panel.

The authors thank the reviewer for pointing out issues. The original purpose for Fig. 4f,g is illustrate that no major conformational flexibility can be detected in the shown Q tunnel loops from Eg-SC I+III₂+IV or Eg-CI under different catalytic conditions. Therefore when super-imposed, they overlay with each other almost perfectly. To avoid confusion as the reviewer has pointed out, we reduced the Fig. 4f,g panels to shown loops from only the Eg-CI under catalytic turnover state. Comparison among Q loops from other conditions are integrated into Fig. S18 panels l-n in a side-by-side fashion for better clarity.

Reviewer #2 (Remarks to the Author):

The authors have answered in an appropriate manner to my comments. I appreciate the modified discussion and the supplemental figures added.

I have no further comments.

Reviewer #3 (Remarks to the Author):

Contributions:

Authors report the high resolution structures of *Euglena* SCs I+III₂+IV and III₂+IV₂. These structures

clarify details about particular features of the Euglena ETC, such as fatty acid synthesis (FAS) and helmet-like domains in complexes I and IV, respectively. Using molecular dynamics simulations (MD), the authors propose a unique supercomplex organization that adapt to the negative membrane curvature which could contribute to fold the discoidal cristae described in this lineage. Also, they cover the active-deactive transition of Euglena CI and the crucial orientation of CIII2 and CIV in their SCs. Finally, based also on MD predictions, authors hypothesize about the binding mechanism of Euglena cyt c to CIV.

The work contributes to expand the knowledge about the structural divergences on ETC complexes among eukaryotic lineages and their consequences on stability and function. I found quite interesting the presented findings, the contribution to the field is clear, expanding the studied lineages, nevertheless some of the conclusions should be clarified or better supported.

Major points

Main text

- A deeper management of the eukaryotic evolution should be reflected as a pillar of the scope, without this, the manuscript focuses only on a detail structural description and gives the impression of been just the report of an “atypical” respirasome structure.

We recognize this issue raised by the reviewer. Indeed the field of mitochondrial bioenergetics diversity is closely related to eukaryotic evolution, however like the many similar publication dissecting the compositions and architectures of the OXPHOS complexes from other eukaryotic supergroups before, the main focus of this manuscript is still structural rather than evolutionary. Nonetheless, we've made quite an effort to address the lack of evolutionary component in manuscript wherever we can. Namely, a chunk of contents have been added on page 6 to discuss the evolutionary rationale for the split of core subunit ND2, which has also been observed for Tetrahymena CI (Zhou et al., 2022). The splitting helps to reduce the overall hydrophobicity of resultant subunit ND2B and ease its transport from cytosol back to mitochondria, as ND2 subunits are no longer included in the streamlined mitochondrial genome of *E. gracilis*. This reflects the evolutionary pressure on the mitochondria-encoded genes to relocate to the nucleus, as demonstrated by the massive gene migrations post-endosymbiosis. On the other hand, the split of hydrophilic core subunit NDUFS1 might relate more to the thermodynamic penalty of folding a very large protein (NDUFS1 is the largest CI subunit) vs folding two smaller subunits after splitting.

We also re-wrote the DISCUSSION section to better incorporate structure-evolution relationship. Specifically, we questioned why so many accessory subunits are gained for ETC complexes post-endosymbiosis without significant gain of novel function outside of electron transport. This is a classic question raised along our earlier report of the Tetrahymena ETC structures (Huynen et al., 2022). Although adaptive narratives can be applied to the universally present accessory domain of γ carbonic anhydrase as well as the FAS domain observed for Eg-CI, their gain are more likely gained non-adaptively through mitonuclear coevolution or due to structural ‘patching’ initiated by mitochondrial gene simplification. This has been demonstrated for the complexification of mito-ribosomes during their evolution. Such theory can well explain the emergence of Eg-specific CIV domains including the matrix domain and the IMS helmet domain.

On the other hand, as Huynen et al. pointed out, adaptive selection cannot be ignored as a driving force for OXPHOS complex complexification. As observed for *Tetrahymena* supercomplex I+III₂, Eg-specific subunits NDUEG7-9 are included as wedges to introduce negative curvature between CI and SC III₂+IV in Eg-respirasome. In this way, Eg-respirasome could better adapt to the discoidal cristae of Euglenozoa. Similar scenario is observed for the specific NDUCA1 NT loop in participation of the novel CI-CIII₂ interface to allow architecture rearrangement and to answer to the adaptive pressure from ETC-cristae coevolution. The above contents have also been added to the DISCUSSION section. In this way, we feel that the updated manuscript is better balanced in terms of structure-evolution relationship without obscuring the focus on structural and mechanistic demonstration of Euglena ETC, therefore better appeals to a broad audience from diverse backgrounds.

- Lines 26-28. A structural FAS domain has been previously identified on isolated Euglena CI (Miranda-Astudillo et al., 2018), additionally, based on complete cell studies the process of anaerobic wax fermentation has been proposed to be linked to CI, electron transfer flavoprotein (ETF), and rhodoquinone (RQ) in Euglena mitochondria (Nakazawa et al., 2018). Nevertheless, no evidences of the catalytic activity of the subunits involved in FAS domain are shown in the present study. Without these evidences, the affirmation put forward on the abstract: “A unique fatty acid synthesis domain locates on the peripheral arm tip of its complex I, linking it to the process of anaerobic wax fermentation” is misleading.

We recognize the work of Miranda-Astudillo et al., 2018, which first identified fatty acid synthesis-related subunits bound on Euglena complex I and also identified the extra domain, proven to be the FAS domain by our results, on tip of Euglena complex I peripheral arm by negative stain EM. Meanwhile, the published works Nakazawa et al., 2018 and Tomiyama et al. 2019 together suggested that acyl-CoA dehydrogenase (ACD) instead of Trans-2-enoyl-CoA reductase (TER) catalysed reaction from enoyl-CoA to acyl-CoA in fatty acid synthesis during Euglena's anaerobic wax fermentation. Usually, ACD functions in β -oxidation, the reversal of fatty acid synthesis, by turning acyl-CoA into enoyl-CoA and depositing electron to electron transfer flavoprotein (ETF) which eventually merged into the respiratory electron transport chain. While the newly identified FAS subunits NDUEG3 and NDUEG5 adopt similar structures to TER, ACD adopts a different fold compared to TER. In this way, the works by Nakazawa and Tomiyama actually argues against the participation of complex I FAS domain in the anaerobic wax fermentation of Euglena.

We tested the TER activity of Euglena's complex I FAS domain, using both NADH and NADPH as electron donors. We also used two typical enoyl-CoA substrates, crotonyl-CoA and (2E)-dodecenoyl-CoA, that we can get commercially. However no significant activity can be detected, compared to the positive control of commercial NADPH-dependent human mitochondrial trans-2-enoyl-CoA reductase (our supplementary figure 16a). Therefore, it is safe to conclude that Euglena's complex I FAS domain lacks the enoyl-CoA reductase activity, let alone its electron supply from ETF or reduced RQ from the Q tunnel of Euglena complex I. Although it is a tempting hypothesis that integrating enoyl-CoA reducing enzyme involved in fatty acid synthesis and its electron source, complex I, into a single protein complex could be kinetically favorable since the two parts are spatially close, it is not supported by current activity data.

Similar situations where ETC complexes gain accessory subunits sharing the structural fold of certain enzyme, but lose its original activity, happen quite frequently in different eukaryotic supergroups. As mentioned in the DISCUSSION section, the most notable example is the γ carbonic anhydrase which is

nearly universally present in all eukaryotic clades but Opisthokonta (Ryan et al, 2010). However, no direct evidence has been reported for its CO₂ to HCO₃⁻ activity as a complex I domain. Other examples include the nucleoside kinase-like subunit NDUFA10 and the NADH-dependent short chain dehydrogenase-like subunit NDUFA9 in opisthokonts' complex I, which both lack respective activities. The rationale for the complication of ETC complexes in 'protists' could be based on constructive neutral evolution theory, or similar to the structural 'patching' process observed for mito-ribosomes (Anton et al, 2019) where the most abundant environmental proteins got associated with large complexes and gradually turns itself into a platform for more structural acquisitions. In this way large complexes like mito-ribosome or respiratory complexes got increasingly diversified in different eukaryotic clades after branching from LECA.

We recognize that considering all the points listed above, the statement 'A unique fatty acid synthesis domain locates on the peripheral arm tip of its complex I, linking it to the process of anaerobic wax fermentation' in the ABSTRACT is not fully grounded. Therefore, this sentence has been revised to 'A unique fatty acid synthesis domain locates on the tip of complex I's peripheral arm, providing a clear picture of its atypical subunit composition identified previously'.

• Lines 253-256. Authors remark a deactivation of Euglena CI and respirasome in presence of N-ethylmaleimide (NEM) similar than the one observed on *Sus scrofa* CI and respirasome but which is not changed upon NADH re-activation. Even though this claim is partially true, the more notable effect, at least comparing CI graphs, is that the deactivation of Euglena CI is a lot less compared with mammalian CI, with this in mind, there is no evident "re-activation" because there is no remarkable "deactivation" to begin with. Authors should rephrase their conclusions.

We respectfully disagree with this point. The thermal deactivation is indeed most significant in mammalian complex I. In mammalian respirasome (Fig. 4b), thermal deactivation is much less compared to mammalian complex I (Fig. 4a) but more similar to Euglena complex I (Fig. 4c). In reviewer's opinion, mammalian respirasome depicted in Fig. 4b does not have a remarkable deactivation, therefore there should not be evident re-activation upon pre-incubation with NADH. However, under each of the three NEM concentrations tested, significant statistical differences exist between activities with or without 10 μM NADH pre-incubation. This is more evident if the bars in Fig. 4a-d are rearranged as shown in the figure below. On the other hand, for Euglena CI (Fig. 4c), no statistical differences exist between activities with or without 10 μM NADH pre-incubation. So it is fair to conclude that although thermal deactivations for mammalian respirasome, Euglena CI and Euglena respirasome are not as evident as mammalian CI, different responses to NADH re-activation can still be distinguished between the two species.

Actually, due to presence of the ferredoxin bridge and the γ carbonic anhydrase domain, levels of complex I thermal deactivation in non-opisthokonts are all less significant compared to mammals. This is exemplified by the recent work of plant supercomplex I+III₂ (Maldonado et al, 2023). Nonetheless, meaningful comparison and discussion about responses of plant complex I deactivation to NADH pre-incubation are still made in this publication. On the other hand, previous work about *Tetrahymena* complex I (Zhou et al, 2022) gave an example of non-deactivatable complex I (also shown in Supplementary fig. 17b and e of the current manuscript), which displayed absolutely no difference upon heating, NEM treatment or NADH pre-incubation. We have made clear in the text that the degree of deactivation for *Euglena* complex I is less than mammalian complex I, although as stated above, this does not invalidate the discussion about different responses to NADH re-activation. Therefore, we believe the current phasing is legitimate.

- The discussion section on main text is only focused on the CI FAS domain biogenesis and the putative explanation of its acquisition on *Euglena* CI. No comments about the role of the helmet domain on CIV are signaled within the main text, some lines should be included about this unique structural feature, whose name comes from a previous work (Miranda-Astudillo et al., 2018), authors should pay attention specially on the cyt c – CIV stable interaction which allowed in the past the demonstration of the full in vitro NADH-O₂ electron transport showed previously on the purified respirasome from this species (Miranda-Astudillo et al., 2021). This is important because previous attempts to measure the in vitro oxidase activity using exogenous cyt c as electron donor revealed a specific requirement of *Euglena* complex IV for its endogenous cyt c (Brönstrup and Hachtel, 1989; Collins et al., 1975). Without these points included into the discussion, the section of the putative binding mechanism of *Euglena* cyt c to CIV based on molecular dynamics is feeble.

The authors recognize the raised point that contents relating to CIV is lacking in the DISCUSSION section. Indeed the term ‘helmet domain’ is inherited from Miranda-Astudillo et al., 2018 which we’ve properly cited as the pioneer work in *Euglena* ETC structures. We’ve added a paragraph about the Eg-specific CIV matrix and IMS helmet domains (page 15) to discuss the evolutionary rationale of their appearance as the mitonuclear coevolution answers the reduction of three TMHs in core subunit COX3 encoded in the mitochondria.

Previous work (Miranda-Astudillo et al., 2021) suggested that native Eg-cyt c can be partially bound to Eg-respirasome after purification process. In our hands, such interaction is not strong enough to ensure NADH:O₂ oxidoreductase activity without addition of external Eg-cyt c to purified Eg-respirasome (see figure below), or allow direct visualization of cyt c density in the respirasome map. This is not surprising, as similar situation is also observed for *Tetrahymena* CIV₂ which has an even more elaborated ‘crater-like’ cyt c binding site compared to the helmet domain-augmented site in Eg-CIV, but still needs addition of recombinant Tt-cyt c to display redox activity and also lacks cyt c density in its cryo-EM map (Zhou et al., 2022, Han et al., 2023). It is worth noting that Tt-CIV₂ also requires endogenous rather than commercial cyt c to function. Therefore, the oxygen consumption by purified Eg-respirasome upon addition of only NADH and quinone could still be debated. On the other hand, visualizing CIV-cyt c interaction structurally via cryo-EM is not a trivial task even in possession of recombinantly expressed Eg-cyt c (see next response). We believe that such investigation would be out of scope of the current manuscript and requires a separate manuscript to be fully addressed.

Supplemental material

- As expected, the molecular dynamics analysis (based on a Eg-cyt c alphafold model) performed twice to elucidate the cyt c – CIV binding site generated two different cyt c binding sites. From these results, authors selected the site which explains better their hypothesis, leaving the other one as a “trap site”, where canonical cyt c could be trapped, explaining the previously observed impossibility of Euglena CIV to reduce exogenous cyt c (Brönstrup and Hachtel, 1989; Collins et al., 1975). In previous works, Euglena respirasome extracted from mitochondrial membranes using GDN detergent (the detergent used in the present manuscript to purify the respirasome by the sucrose gradient) could be purified with the endogenous cyt c attached as could be corroborated by spectroscopy and the measure of full NADH – O₂ electron flow (Miranda-Astudillo et al., 2021). These indicates a strong and stable binding property, which is not observed in the presented manuscript. Authors should take advantage on the technological tasks presented in their manuscript, and a reconstitution between the purified CIV (or any SC I/III₂/IV or III₂/IV₂) and the already purified recombinant Eg-cyt c should be performed in a similar way as previously showed with the bovine oxidase and horse cyt c (Shimada et al., 2017) to elucidate the correct binding mechanism. Without these proper analysis, the proposed CIV-cyt c mechanism is speculative.

We appreciate the valuable feedback from the reviewer and recognize their concerns regarding the statistical robustness of our MD results. To address these concerns, we performed two additional repeat simulations (Run 2 and 3), extending each to 2000 nanoseconds. These supplementary simulations showed cyt c binding patterns consistent with our initial observations (Run 1), lending support to the reliability of the proposed binding site, though some minor differences were noted in the conformational poses of cyt c. The trajectories from all these simulations have been made publicly available on GitHub (<https://github.com/yongwangCPH/papers/tree/main/2023/CIV-cytC>). Despite these promising results, we recognize the inherent limitations of MD simulations due to their confined timescales, which constrain our ability to accurately quantify binding affinities and rank binding site stability. As such, we have revised the manuscript to state: "Although our simulations offer a feasible model for the binding mechanism, we acknowledge the difficulties in precisely quantifying binding affinities and comparing binding site stabilities given the limited timescales achievable with MD and the probable inadequate conformational sampling. These limitations present opportunities for further investigation to build upon these preliminary findings."

As mentioned in the last response, we cannot reliably reproduce the partial co-purification of natively bound Eg-cyt c by Eg-respirasome, both in terms of activity measurements and in terms of direct visualization of cyt c density in the cryo-EM map of Eg-respirasome. Generally speaking, as pointed out

by Shimada et al., 2017 interaction between CIV and cyt c could not be too strong, otherwise the fast-on, fast-off property of cyt c as a soluble electron carrier could not be guaranteed.

Theoretically, one could add a large excess of recombinant Eg-cyt c to purified Eg-respirasome and perform routine cryo-EM based structural analysis of the mixture, in the hope that cyt c binding sites in CIV are saturated by an overdose of cyt c. Such experiment has already been performed for yeast supercomplex III₂+IV (Moe et al, 2021 PNAS, Moe et al, 2023 PNAS). Close inspection of the obtained cryo-EM maps containing a molar ratio 12:1 overdose of cyt c indicates that only very weak to none (not distinguishable from noise) density could be observed in the raw cryo-EM maps. Therefore, it is logical to argue that even if similar experiment is repeated for *Euglena*, not much conclusive mechanistic insight can be extracted regarding Eg-CIV-cyt c interaction. Actually, both existing structures of cyt c bound CIII₂ and CIV (Lange et al, 2002, PNAS, Shimada et al., 2017) are crystallisation but cryo-EM based.

We recognise that investigation into CIV-cyt c interaction is of significance in the field of bioenergetics, however, it is out of the scope of the current manuscript. Structural study of the interaction mode between Eg-CIV and cyt c via crystallography is most valuable but belongs to the content of a separate manuscript. Here we raise our concerns regarding this point and hope that the reviewer could further consider upon the balance between the comprehensiveness and the focus of a published manuscript.

Minor points

Main text

- lines 40 and 41. “IV₂+I+III₂+II” and “IV₂+(I+III₂+II)” represent the same SC, check stoichiometry. It’s a typo, should be “MC IV₂+(I+III₂+II)₂” and has been corrected in the text

- line 51. Remove the term “protist”, nowadays this term is obsolete, and the recent evidence showed that the text-book kingdoms, e.g. animals, plants, fungi and protist, does not reflect the eukaryotic diversity and complexity (Burki et al., 2020; Keeling and Burki, 2019).

The authors thank the reviewer for pointing out this inaccuracy. The text has been changed to ‘*Euglena gracilis* is a free-living, flagellated single cell eukaryote...’

- lines 53 and 370. Recent studies addressed the split of the former Excavata supergroup into new supergroups, Euglenoids should be included into the Discoba supergroup (Burki et al., 2020).

The authors thank the reviewer for pointing out this inaccuracy. The usage of term ‘Excavata’ has been substituted by ‘Discoba’.

- line 56. It should be important to clarify the secondary origin of the plastid in this species, because after each endosymbiotic process massive gene migrations were performed, contributing to the genetic wealth present in these domains (Leister, 2005).

The authors thank the reviewer for pointing out this inaccuracy. The text has been revised.

- line 66. It reads: “to fill the missing gap...”, it should be read: “to fill a missing gap...”, even though this manuscript contributes to expand our knowledge among bioenergetics of eukaryotic organisms, there are still a lot of missing information related to other eukaryotic supergroups.

The authors thank the reviewer for pointing out this phasing issue. The text has been revised.

- line 191. A comparison of *Euglena* respirasome against alveolate (PDB 7TGH) respirasome is indicated, but in the corresponding figure 3(c), only *O. aries*, and *A. thaliana* are visible alongside the *Euglena* structure.

The authors thank the reviewer for pointing out this inaccuracy. In Fig 3c, *E. gracilis* respirasome is compared to its mammalian and plant counterparts, while in Fig 3d, the comparison is done among *E. gracilis*, mammalian and *T. thermophila* respirasome/SC I+III₂. The legend for Fig. 3 is revised to clarify this point. The sentence at line 191 serves as a general introduction to all the structural comparisons performed in the text below, therefore all published models used in Fig. 3 are mentioned here. The figure citation here are revised from ‘Fig.3 a,b’ to ‘Fig. 3’ as a whole to better clarify this point. In the text below, panels Fig. 3c and Fig. 3d are individually cited to represent structural comparison involving *A. thaliana* or *T. thermophila* supercomplexes. Fig. 3d, which includes comparison with *T. thermophila* SC I+III₂, is cited at the end of the 2nd paragraph of this section.

- line 715. Reference 55 listed on “Methods references” is not related to the heterotrophic culture of *Euglena gracilis*, and the conditions described in lines 713 – 715 resemble to previous reported conditions (Yadav et al., 2017). Please correct this reference.

The authors thank the reviewer for pointing out this issue. It seems that the numbering of the method references have been messed up when inserting new citations in the main text during the last revision. The authors meant to cite the reference (Buetow et al., 1963) here. The method reference numbering has been corrected and the (Yadav et al., 2017) citation has been added for *E. gracilis* culture.

- Figure 2 (g-i) For clarity, please indicate if there is any rotation of the “point of view” from the corresponding images.

The authors thank the reviewer for pointing out this issue. There are ~90 degree rotation of the ‘point of view’ between panels g and h, as well as between panels h and i. These are now labeled in Figure 2 and also clarified in the legend.

- Figure 3 (c) *T. thermophila* (PDB 7TGH) SC is labeled in the figure legend, but is not visible in the corresponding panel.

This comment points out the same issue with the comment regarding line 191 above. The legend for Fig. 3 and the figure citation in the main text have been revised to address this issue.

REVIEWERS' COMMENTS

Reviewer #3 (Remarks to the Author):

In this revised version of the manuscript Authors have made an effort to answer all the points raised in the previous revisions, now they have addressed the lack of evolutionary point of view of the obtained data, and the inclusion of this vision enriches the overall manuscript and makes it more attractive for a global audience.

Additionally, most of the changes made in the figures and main text contribute to make it easy to follow the entire manuscript to the non-structural-related audience.

One quite interesting and positive addition is the new Supplementary Tables 6-8 which reflect the polypeptide diversity among the different linages, pillar of the structural variety observed in the eukaryotic structures described so far.

The authors have addressed correctly my previous comments, and performed the majority of the suggested changes accordingly. I have no further comments to the work.

Only one additional change is missing in this version: Suppl. Information, Line 46, is better to name "Discoba (former part of Excavata clade)" instead of "Excavata" to be in agreement with the changes made in the main text.

Reviewer #3 (Remarks to the Author):

In this revised version of the manuscript Authors have made an effort to answer all the points raised in the previous revisions, now they have addressed the lack of evolutionary point of view of the obtained data, and the inclusion of this vision enriches the overall manuscript and makes it more attractive for a global audience.

Additionally, most of the changes made in the figures and main text contribute to make it easy to follow the entire manuscript to the non-structural-related audience.

One quite interesting and positive addition is the new Supplementary Tables 6-8 which reflect the polypeptide diversity among the different linages, pillar of the structural variety observed in the eukaryotic structures described so far.

The authors have addressed correctly my previous comments, and performed the majority of the suggested changes accordingly. I have no further comments to the work.

Only one additional change is missing in this version: Suppl. Information, Line 46, is better to name “Discoba (former part of Excavata clade)” instead of “Excavata” to be in agreement with the changes made in the main text.

The authors thank the reviewer for the recognition of our revised manuscript as well as for pointing out the error. The “Excavata” in the supplementary discussion has been revised to “Discoba (former part of Excavata clade)” according to the reviewer’s instructions.